

# Coastal HF radars in the Mediterranean: status of operations and a framework for future development

Pablo Lorente[1,2], Eva Aguiar[3], Michele Bendoni[4], Maristella Berta[5], Carlo Brandini[4,6], Alejandro Cáceres-Euse[7], Fulvio Capodici[8], Daniella Cianelli[9,10], Giuseppe Ciraolo[8], Lorenzo Corgnati[5], Vlado Dadić[11], Bartolomeo Doronzo[4,6], Aldo Drago[12], Dylan Dumas[7], Pierpaolo Falco[13], Maria Fattorini[4,6], Adam Gauci[12], Roberto Gómez[14], Annalisa Griffa[5], Charles-Antoine Guérin[7], Ismael Hernández-Carrasco[15], Jaime Hernández-Lasheras[3], Matjaz Ličer[16,17], Marcelo Magaldi[5], Carlo Mantovani[5], Hrvoje Mihanović[11], Anne Molcard[7], Baptiste Mourre[3], Alejandro Orfila[15], Adèle Révelard[3], Emma Reyes[3], Jorge Sánchez[18], Simona Saviano[9,10], Roberta Sciascia[5], Stefano Taddei[4], Joaquín Tintoré[3,15], Yaron Toledo[19], Laura Ursella[20], Marco Uttieri[9,10], Ivica Vilibić[11,21], Enrico Zambianchi[10,22], Vanessa Cardin[20]

[1]Puertos del Estado, Madrid, 28042, Spain
[2]NOLOGIN CONSULTING SL, Zaragoza, 50018, Spain.
[3]SOCIB -Balearic Islands Coastal Ocean Observing and Forecasting System-, Palma, 07122, Spain.
[4]Consorzio LaMMA, Livorno, 57126, Italy.
[5]Consiglio Nazionale delle Ricerche (CNR), Istituto di Scienze Marine (ISMAR), Lerici, 19032, Italy.
[6]Consiglio Nazionale delle Ricerche (CNR), Istituto per la Bioeconomia (IBE), Sesto Fiorentino, 50019, Italy.
[7]Mediterranean Institute of Oceanography, Université de Toulon, Aix Marseille Univ, CNRS, IRD, MIO, Toulon, 83041, France
[8]Università degli Studi di Palermo, Palermo, 90218, Italy.
[9]Stazione Zoologica Anton Dohrn, Naples, 80121, Italy.
[10]Consorzio Nazionale Interuniversitario per le Scienze del Mare (CoNISMa), Rome, 00196, Italy.
[11]Institute of Oceanography and Fisheries, Split, 21000, Croatia.
[12]Physical Oceanography Research Group, University of Malta, Msida, MSD 2080, Malta.
[13]Universita' Politecnica delle Marche, DISVA, Ancona, 60121, Italy.
[14]Helzel Messtechnik GmbH, Kaltenkirchen, 24568, Germany.
[15]Mediterranean Institute for Advanced Studies – IMEDEA- (CSIC-UIB), Esporles, 07190, Spain.
[16]National Institute of Biology, Marine Biology Station, Piran, 6330, Slovenia.
[17]Slovenian Environment Agency, Ljubljana, SI-1000, Slovenia.
[18]Qualitas Instruments S.A., Madrid, 28043, Spain.
[19]School of Mechanical Engineering, Tel-Aviv University, Tel-Aviv, 6905904, Israel.
[20]Istituto Nazionale di Oceanografia e di Geofisica Sperimentale, OGS, Sgonico TS, 34010, Italy.
[21]Ruđer Bošković Institute, Division for Marine and Environmental Research, Zagreb, 10000, Croatia.
[22]Dipartimento di Scienze e Tecnologie (DiST), Parthenope University of Naples, Naples, 80143, Italy.

*Correspondence to*: Pablo Lorente (plorente_externo@puertos.es)

**Abstract.** Due to the semi-enclosed nature of the Mediterranean Sea, natural disasters and anthropogenic activities impose stronger pressures on its coastal ecosystems than in any other sea of the world. With the aim of responding adequately to science priorities and societal challenges, littoral waters must be effectively monitored with High-Frequency radar (HFR)





systems. This land-based remote sensing technology can provide, in near real-time, fine-resolution maps of the surface circulation over broad coastal areas, along with reliable directional wave and wind information. The main goal of this work is to showcase the current status of the Mediterranean HFR network and the future roadmap for orchestrated actions. Ongoing collaborative efforts and recent progress of this regional alliance are not only described but also connected with
other European initiatives and global frameworks, highlighting the advantages of this cost-effective instrument for the multi-parameter monitoring of the sea state. Coordinated endeavours between HFR operators from different multi-disciplinary institutions are mandatory to reach a mature stage at both national and regional levels, striving to: i) harmonize deployment and maintenance practices; ii) standardize data, metadata and quality control procedures; iii) centralize data management, visualization and access platforms; iv) develop practical applications of societal benefit, that can be used for strategic
planning and informed decision-making in the Mediterranean marine environment. Such fit-for-purpose applications can serve for search and rescue operations, safe vessel navigation, tracking of marine pollutants, the monitoring of extreme events or the investigation of transport processes and the connectivity between offshore waters and coastal ecosystems. Finally, future prospects within the Mediterranean framework are discussed along with a wealth of socio-economic, technical and scientific challenges to be faced during the implementation of this integrated HFR regional network.

## 55  1 The Mediterranean Sea coastal regions: science priorities and societal needs

### 1.1 The oceanographic landscape

The Mediterranean Sea is located at the crossroads of three continents (Africa, Europe and Asia), thereby playing an important geopolitical role in the world chessboard since ancient times as a restless navigable route for maritime transport, commerce and cultural exchange (Gaiser and Hribar, 2012). It is a semi-enclosed, microtidal basin connected to the Atlantic
Ocean, the Black Sea and the Red Sea by three geostrategic chokepoints such as the Strait of Gibraltar (in the west), the Dardanelles (in the northeast) and the Suez Canal (in the southeast), respectively (Fig. 1). It is also an oligotrophic well-oxygenated system, characterized by complex physical and biological dynamics (Christaki et al., 2011). Offshore waters exhibit extremely low biological productivity, with the concentration of nutrients decreasing from NW to SE. The panoramic picture of the Mediterranean circulation, which exhibits a strong seasonal and inter-annual variability, is composed by a
variety of relevant processes interacting at diverse timescales, namely: water mass formation, overturning circulation, boundary currents and frontal instabilities (Pinardi et al., 2019; Tintoré et al., 2019). The large-scale thermohaline circulation is interconnected with recurrent sub-basin gyres and energetic mesoscale eddies, which are in turn bounded by current meanders and bifurcating jets (Millot and Taupier-Letage, 2005). The rugged configuration of narrow shelf areas, with steep continental breaks, entails the intrusion and direct impact of the large-scale open ocean flow on the coastal dynamics. For
further details about general oceanographic conditions in the entire basin, the reader is referred to Pinardi et al. (2006) and Malanotte-Rizzoli et al. (2014).



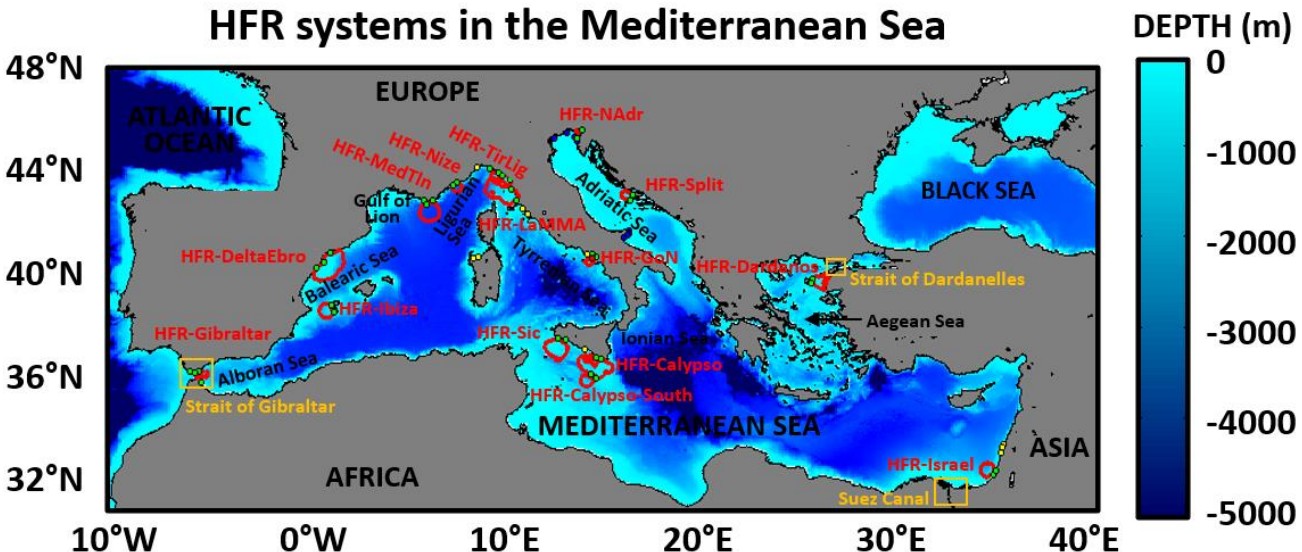

**Figure 1. Bathymetric map of the Mediterranean Sea, depicting some local seas and geographical features. The location and spatial coverage of ongoing High Frequency radar (HFR) systems deployed in the basin is represented with red contours. Ongoing, old and future HFR sites are represented with green, blue and yellow dots, respectively.**

## 1.2 Science priorities

The Mediterranean Sea is one of the biggest reservoirs of marine life in the world, contributing to more than 7% of world's marine biodiversity including a high percentage of endemic species (Coll et al., 2010). Since natural disasters, anthropogenic activities and climate change might impose significant and long-lasting pressures (Juza and Tintoré, 2021; Tuel and Eltahir, 2020; Spalding et al., 2014), diverse science priorities have been identified to promote healthy and sustainable marine ecosystems in the Mediterranean Sea, among others:

i)  The detailed investigation of transport processes and the connectivity between offshore waters and coastal ecosystems. The cross-shelf exchange of nutrients, organic matter, pollutants and other passive tracers might have relevant implications in terms of intense biogeochemical activity, eutrophication, proliferation of harmful algal blooms and fisheries production. Equally, a deeper understanding of the ocean circulation can lead to more accurate model predictions of Lagrangian trajectories which in turn can be used to gain insight into particle tracking, dispersion processes, residence times and water renewal mechanisms.

ii)  The impact assessment of coastal hazards and extreme sea states, ranging from storm surges, erosion and flash-floods to rogue waves and the Mediterranean hurricanes, also named "Medicanes" (Von Schuckmann et al., 2020; Milglietta and Rotunno, 2019; Wolff et al., 2018; Cavaleri et al, 2012).



iii) The thorough analysis of climate-driven variations such as sea level rise, the steady acidification, the increase of ocean
heat content, recurrent marine heat waves or potential alterations in the thermohaline circulation (Juza and Tintoré, 2021;
Garrabou et al., 2019).

## 1.3 Societal challenges

The aforementioned science priorities are particularly motivated by the semi-enclosed nature of the Mediterranean Sea,
where anthropogenic pressures are likely more intense than in any other sea of the world (Lejeusne et al. 2010). An
increasingly high density of inhabitants (above 470 millions) gravitate for their living needs in littoral regions (Wolff et al.,
2020), which are not only impacted by local human activities but also further altered by massive international tourism,
including passenger ferries, cruises and recreational boating. Apart from the shortage of water resources (tied to the
population growth and the intensification of coastal urbanization, agricultural development and industrial activities), other
interconnected societal challenges in the Mediterranean Sea include:

i)   Enhanced maritime safety. An efficient ship routing is required to minimize both fuel consumption and the risk of
accidental oil spills. Furthermore, search and rescue (SAR) operations constitute a major humanitarian emergency in the
Mediterranean basin and thereby demand science-based management protocols for a timely response.

ii)  Improved ecological decision support systems. The preservation of local marine fauna (seriously jeopardized by intense
overfishing), the habitat modification, the transfer of alien species or the ingestion of litter demand tailored tools for
informed decision-making (Campanale et al., 2019). Equally, the monitoring of water quality in the Mediterranean Sea
remains as a priority, since it is negatively impacted by the discharge of land-based toxic pollutants from local rivers into
coastal sea waters (Nikolaidis et al., 2014) and also by episodic marine pollution episodes (Soussi et al., 2020).

## 1.4 Multi-platform observing systems: High Frequency radar as a key component

To adequately respond to those science priorities and societal challenges previously enumerated, a sustainable multi-
platform observing infrastructure should be implemented and integrated. The accurate monitoring and deep understanding of
the Mediterranean marine environment are not only crucial to prompt a wealth of anticipatory adaptation strategies but also
of great economic value for the maritime sector (Melet et al., 2020). Such preventive approaches can aid to bridge the gap
between marine citizen science and coastal management (Turicchia et al., 2021), which would strengthen the community
resilience at multiple scales (Summers et al., 2018; Linnenluecke et al., 2012).

With the advent of new technologies and ships supporting interdisciplinary suites of sensors (Mahadevan et al., 2020), a
growing wealth of observational data are nowadays available to properly characterize the Mediterranean Sea (Le Traon et al.,
2013). Most of these data are regularly ingested by the Copernicus Marine Environment Monitoring Service In Situ
Thematic Center, hereinafter CMEMS-INSTAC (Le Traon et al., 2017), the EMODnet programme (Martín Miguez et al.,
2019) or the SeaDataCloud (Fichaut and Schaap, 2016), promoting the ocean observing value chain that links observations
and data discovery to downstream applications of societal benefit.





For instance, novel satellite missions such as the Soil Moisture and Ocean Salinity (SMOS) or the Surface Water and Ocean Topography (SWOT) aim to increase the resolution capacity in the coastal band to properly feature, respectively, the salinity field (Olmedo et al., 2018) and the submesoscale circulation (Gómez-Navarro et al., 2018). Arrays of ARGO profiling floats, which provide temperature and salinity measurements down to 2000 m (Kassis and Korres, 2020; Sánchez-Román et al., 2017), are nowadays extended to the deep ocean and further complemented with data from biogeochemical Argo

(D'Ortenzio et al. 2020) and bio-physical gliders (Cotroneo et al., 2019; Barceló-Llull et al., 2019).

In situ measurements provided by conventional instruments such as point-wise current meters (PCM), Acoustic Doppler Current Profilers (ADCPs) or drifting buoys (Sotillo et al., 2016) are useful to monitor the Mediterranean circulation, but present some limitations in terms of spatial resolution and areal coverage. A complementary and relatively novel technology that has been steadily gaining worldwide recognition as an effective shore-based remote sensing instrument is high

frequency radar (HFR). HFR networks have become an essential component of coastal ocean observation since they collect, in near real time, fine-resolution maps of the surface circulation over broad coastal areas, providing thereby a dynamical framework for other traditional in-situ observation platforms (Roarty et al., 2019, Rubio et al., 2017). They provide two-dimensional synoptic maps of surface currents for distances up to 200 km offshore over a wide variety of high spatial (0.2-6 km) and temporal (usually between 15-minute and 1-hour averages) scales, enabling the detailed monitoring of

(sub)mesoscale coastal processes. Although HFR-derived wave and wind measurements are not yet seen as operational products, there are many publications that demonstrate these capabilities (Esposito et al., 2018; Wyatt, 2006 and 2018).

Additionally, HFR data present a broad range of science-based applications of societal benefit, such as maritime security (safe vessel navigation and SAR operations), tracking the dispersion and retention of marine pollutants (oil spill mitigation), effective monitoring of extreme events, fisheries and coastal management (e.g. port activity and impact on marine protected

areas). Other emerging uses include vessel tracking, ocean energy production, or even tsunami detection (Roarty et al., 2013; Lipa et al., 2012).

Finally, it is worthwhile mentioning that a combined use of multi-platform observing systems, encompassing both in situ (buoys, ADCPs, drifters, tide gauges, etc.) and remote (HFRs, altimetry products, etc.) sensors, can provide additional insight into the comprehensive three-dimensional characterization of the Mediterranean Sea state at multiple scales. Equally,

it can also contribute positively to a more exhaustive skill assessment of hydrodynamic, biogeochemical and wave forecast systems running operationally in this regional basin (Aguiar et al., 2020; Mourre et al., 2018, Lorente et al., 2016a, 2016b and 2019). The implementation of consistent data assimilation schemes has constituted a quantum leap in terms of realistic forecast predictions in the Mediterranean Sea since they maximize the interconnection of ocean observing systems and numerical models (Teruzzi et al., 2018; Dobricic and Pinardi, 2008). In this context, the benefits of assimilating HFR current

data to improve ocean model forecasts in the Mediterranean region have been also demonstrated (Hernández-Lasheras et al., 2021; Vandenbulcke et al., 2017; Marmain et al., 2014).



## 1.5 The Mediterranean oceanographic network

The Mediterranean Oceanographic Network (www.mongoos.eu), together with EuroGOOS, is part of the 13 Global Regional Alliances of the Global Ocean Observing System (GOOS) that aims to develop both sustained ocean monitoring
and tailored services to meet regional and national priorities, aligning the global goals of GOOS (www.goosocean.org) with the implementation of fit-for-purpose applications to satisfy local requirements (Moltmann et al., 2019). At European level, MONGOOS plays a key role as one of the five Regional Operational Oceanographic Systems (ROOS) of EuroGOOS, aiding to bridge the gap between the north (Europe) and the south (Africa) shores of the Mediterranean Sea.

It was established in 2012 as a collaborative framework to further develop operational oceanography and sustained
observations collection in the Mediterranean Sea. The network, based on its scientific and strategic plan (Sarantis et al., 2018), boots a science-oriented vision as well as technological developments, necessary to efficiently promote regional monitoring capabilities in the Mediterranean area.

MONGOOS engages in activities related to scientific promotion, the fostering of applications for societal benefits, and the production and use of operational oceanography services. Its science and strategy plan is fully aligned with the BlueMED
implementation plan (Fig. 2), where the establishment of a fully-integrated multi-platform monitoring system was acknowledged as crucial to develop a sustainable Blue Economy in the Mediterranean area (Trincardi et al., 2020). Furthermore, it is also in line with the EU-2020 Green Deal call named "Digital Twin of the Ocean". It consists of the integration of existing leading-edge capacities in ocean observation and forecasting with top-tier digital technologies (cloud infrastructures, supercomputing resources, artificial intelligence, etc.) to adequately provide a high-resolution, three-
dimensional description of the ocean state in near real time.

MONGOOS also contributes to the Decade of Ocean Science for sustainable development (2021-2030) initiative, which was proclaimed by the United Nations and relies on sustained ocean observations. It aims to create partnerships, strengthen international cooperation, mobilize resources, engage governments (and targeted stakeholders) and support high-stakes decision-making in the marine environment (Ryabinin et al., 2019). The network plays an important role in "The Science We
Need for the Mediterranean Sea We Want" Programme (SciNMeet) recently endorsed in the first Call for Decade Action and which encompasses a broad scope and high ambition to tackle all major environmental and social challenges in the Mediterranean basin (Fig. 2).

The MONGOOS network is formed by three working groups in charge of fostering the activity in specific areas, namely: Observation, Modelling and Application working groups. The Mediterranean HFR network, participated by 7 countries
(Israel, Croatia, Slovenia, Malta, Italy, France, and Spain), has become an essential component of the Mediterranean observing system. These infrastructures are key elements for Coastal Observing Systems providing near real time ocean currents with direct implications in monitoring large (regional) areas. Present applications include: i) maritime safety; ii) extreme hazards and iii) environmental transport processes which will be reviewed in a companion paper.



**Figure 2. Conceptual framework for ocean observing systems, alliances and initiatives, ranging from global to regional scales. OceanOPS, which depends both on the UNESCO Intergovernmental Oceanographic Commission and the World Meteorological Organization, represents the operational centre of GOOS where meteo-oceanographic observing systems are centralized.**

## 1.6 Objectives of the work

Motivated by the increasing relevance of the consolidated HFR technology, this work pursues several interrelated goals:



i) Showcase the current status of the Mediterranean HFR network, providing a succinct description of each HFR system. Ongoing work plans, recent progress in basic products and applications are enumerated, thereby highlighting the benefits of this cost-effective technology for the multi-parameter monitoring of coastal waters.

ii) Show the links of this HFR network with diverse multi-institutional initiatives and alliances at regional and global level, emphasizing the bidirectional interactions with the Global HFR network (Roarty et al., 2019), the HFR EuroGOOS task Team (Rubio et al., 2017), GOOS and EuroGOOS (Fig. 2). Equally, the connections with other European initiatives such as the Copernicus Marine Environment Monitoring Service -CMEMS- (Le Traon et al., 2017) and cross-border projects (e.g. EuroSea, Jerico-Next, Impact, Sicomar, Sinapsi, CALYPSO, etc.) are also presented.

iii) Delineate future prospects within the Mediterranean framework along with the number of challenges to be faced, encompassing economic, technical and scientific aspects.

This manuscript, which constitutes the first part of a double contribution, aims to provide a panoramic overview of the roadmap to transform individual HFR systems into a fully integrated, mature network operated permanently in the Mediterranean Sea. The second part focuses on the latest scientific breakthroughs and diverse research-based applications of HFR data, fully aligned with pre-defined science priorities, in order to meet both societal needs and stakeholders demands in an innovative way (Reyes et al., submitted to this Special Issue).

The paper is organized as follows: Section 2 describes not only the fundamentals of HFR technology but also basic products, encompassing the retrieval of surface currents, wave parameters and directional wind estimations. Sections 3 and 4 outline fundamental technical aspects of each HFR system within this regional network and a number of collaborative projects, respectively. Ongoing and future challenges to be faced over the next decade are discussed in Section 5. Finally, main conclusions are drawn in Section 6.

## 2. HFR systems in MONGOOS: valuable assets for operational coastal oceanography

### 2.1 Fundamentals of HFR technology

The HFR technology, founded on the principle of Bragg scattering of the electromagnetic radiation over the rough conductive sea surface (Crombie, 1955), infers the radial current component from the Doppler shift of radio waves backscattered by surface gravity waves of half their electromagnetic wavelength. Each single radar site is configured to estimate radial currents moving toward or away from the receive antenna. Since the speed of the wave is easily derived from linear wave theory, the velocity of the underlying ocean surface currents can be computed by subtraction. The distance to the backscattered signal is determined by range-gating the returns. Although all HFR systems rely on fundamentally similar physics and Doppler processing algorithms to infer the range and radial velocity of the scattering surface, they differ in the reception and interpretation of the incoming direction of the backscattered signal.

According to the methodology used to determine the incoming direction of the scattered signal (also named "bearing determination"), commercial HFR systems can be differentiated into two major types: Beam Forming (BF) and Direction



Finding (DF). BF radars use linear phased arrays of receive antennas (typically between 8 to 16 antennas in a linear array) to electronically point towards a sector of ocean surface, which amplifies signal strength from that direction and attenuates
signal from other directions. The WEllen RAdar (WERA), developed by the University of Hamburg and manufactured by Helzel Messtechnik GmbH (Gurgel et al., 1999), is one example of BF radar. DF radars, such as the Coastal Ocean Dynamics Application Radar (CODAR) SeaSonde (Barrick et al., 1985), measure the return signal continuously over all angles, exploiting the directional properties of a three-element antenna system (two directionally dependent orthogonal crossed loops and a single omnidirectional monopole) and use the Multiple Signal Characterization (MUSIC) DF algorithm
(Schmidt, 1986) in order to determine the direction of the incoming signals.

A large number of HFR systems are active worldwide operating at specific frequencies within the 3–30 MHz band and providing radial measurements which are representative of current velocities in the upper 0.5–2 m of the water column. In regions of overlapping coverage from two or more sites, radial current estimations are geometrically combined to estimate total current vectors on a predefined Cartesian regular grid. The specific geometry of the HFR domain and, hence, the
intersection angles of radial vectors influence on the accuracy of the total current vectors resolved at each grid point. Such a source of uncertainty is quantified by a dimensionless parameter denominated Geometrical Dilution of Precision -GDOP- (Chapman and Graber, 1997), which typically increases with the distance from the HFR sites.

Another relevant difference among HFR systems is the way the signal is transmitted and received. Typically, HFRs transmit using Frequency Modulated Continuous Wave (FMCW) which consist of a signal whose frequency is linearly swept (also
called chirp signal). Using pure FMCW, the transmitter and receiver antennas are constantly transmitting and receiving. This means that the receiver antenna has to be physically separated from the transmitter to reduce direct leak of the transmitted signal into the receivers, which may saturate the electronics. Compact versions of HFR implement interrupted FMCW (iFMCW or FMiCW) in which the transmit signal is switched off and on repetitively. Under this scheme, the receivers process backscattered information from the off-state of transmission only. This improves the isolation of direct leakage of
transmitted signal into the receivers, enabling very compact antenna configurations where transmit and receive antenna are collocated, and usage of the same antenna both as transmitter and receiver. Some phased-array versions of HFR also implement FMiCW to avoid saturation of the receiver's ADC from the strong transmitted signal, which could deteriorate the correct measurement of the signals coming from the ocean if the adequate separation between transmitter and receiver is not taken into account.

Due to the lack of interruption on the receiver, pure FMCW harnesses more backscatter energy from the ocean improving the range performance of the HFRs (Heron et al., 2015). Also, the type of processing impacts the integration time which is usually higher with DF than BF. The reason is that DF processing requires a sufficient number of sample Doppler spectra (hence a longer integration time) to estimate the covariance matrix which is at the heart of the method. Less integration time can be advantageous for extremely dynamic seas or for specific applications such as tsunami detection and ship tracking.
However, for both FMCW and FMiCW (either BF or DF), reducing the integration time results in less accurate surface currents outputs as averaging measurements at different levels might get rid of: i) chaotic changes due to turbulence; ii)



subgrid scale variability of the surface current; and iii) random fluctuations of the sea echo itself due to the Gaussian nature of the Bragg ocean waves and the linear transformation represented by the scattering from them (Barrick, 1980; Wang et al., 2014).

Phased-array systems can also employ DF techniques. The WERA system is available as well in a configuration using a squared receive array of 4 antennas (not collocated) which employs DF techniques, although this option is not widely used. The only example in the Mediterranean Sea with such a configuration is given in Zervakis (2017). DF techniques have also been applied to linear arrays, improving further the azimuth resolution (Barbin et al., 2009 and 2011). More recently, some operational applications have been developed in the Mediterranean Sea by using a hybrid approach that applies both BF

(antenna grouping) and DF techniques on phased-array HFR systems (Dumas et al., 2020).

   Robust surface current measurements can be derived from the Doppler shift of the dominant first-order peak in the radar echo spectrum (Stewart and Joy, 1974). The use of first-order peaks to measure wind direction, albeit less explored, has been previously reported in the literature (Heron, 2002; Lipa et al., 2014; Kirincich, 2016; Hisaki, 2017; Shen & Gurgel, 2018; Wyatt, 2018; Saviano, 2021). The directional wave spectrum and derived parameters such as local significant wave height,

centroid wave period and mean wave direction can be determined from the weaker second-order sea-echo Doppler spectrum by adopting two main approaches: full integral inversion or fitting with a model of ocean wave spectrum (Lipa and Nyden 2005). A variety of inverse techniques have been developed over the last years (Barrick, 1977; Wyatt, 1990; Hisaki, 2006).

   The second-order scattering-based methods significantly rely on the echo quality which varies with sea state and radar frequency (Wyatt et al., 2005). The relative contribution of the second-order spectrum increases with both the radar

frequency and the wave height. Since wave data are dependent upon the occurrence of both Bragg and larger surface gravity waves, there is a minimum threshold for sea states at a given radar frequency in which reliable wave parameters can be determined. Below such sensitivity threshold, the lower-energy second-order spectrum is closer to the noise floor and more likely to be contaminated with spurious contributions that might result in wave height overestimation or limited temporal continuity in wave measurements (Lipa and Nyden, 2005; Tian et al., 2017). During extreme weather events, there is also a

limiting factor for HFR accuracy as the wave height increases and exceeds the saturation limit defined (on an inverse proportion) by the radar transmit frequency. If the radar spectrum saturates, the first-order peak merges with the second-order one and interpretation of the spectra becomes impossible with existing methods (Forney, Roarty, and Glenn, 2015).

   The development of robust validation methodologies constitutes a core activity when implementing a fully operational network since HFR measurements might be subject to some error sources and potential uncertainties. Inherent problems of

HFR technology, such as power-line disturbances, radio frequency interferences, ionosphere clutter, environmental noise, unresolved velocity fluctuations, reflections from moving ships, off-shore wind turbine interferences, adverse environmental conditions, improper determination of the angle of arrival, limitations in signal processing methods, antenna pattern distortions or hardware failures likely impact on the accuracy of HFR measurements (Paduan et al., 2006; Kohut and Glenn, 2003). Since HFR is gaining ever-wider acceptance by the oceanographic community as an efficient land-based technology

for the multi-parameter monitoring of the sea state in near real-time, it is essential to routinely assess the accuracy of HFR





measurements against independent in situ observations, fostering the subsequent use for research purposes and the development of added-value operational tools.

## 2.2 Basic HFR products

### 2.2.1 High-Frequency radar surface current monitoring, improvement and validation

The primary goal of oceanographic HFR systems is the derivation of radial and total ocean surface currents from the backscattered signal on the receiving antennas. The measurement principle relies on the first-order "Bragg theory" according to which the dominant contribution to the backscattered electromagnetic field is the resonant surface wave with half radar-wavelength. This results in a couple of sharp peaks in the positive and negative part of the Doppler spectrum located at the so-called "Bragg frequency" and its opposite. This remarkable property was first experimentally observed by (Crombie,

1955) and given a solid theoretical framework by Barrick (1972). It was later realized that this could be used to infer the radial surface current by measuring the frequency shift between the theoretical and observed positions of the Bragg peaks (Stewart and Joy 1974).

Despite the simplicity of the physical concept, the estimation of the radial current in every sea surface patch from the mere antenna voltage requires a chain of technical and processing steps which are far from being trivial. It also implies choices

and compromises from the operator depending on the logistical constraints and the aimed applications. The main limitation is the ability to properly identify the first-order Bragg peaks and their exact locations. This is related to the signal-to-noise ratio (SNR), the integration time and the number of sample spectra which are combined over the observation time. The SNR is primarily dependent on the transmitted power and determines the effective range of the HFR. Increasing the integration time improves both the SNR and the Doppler resolution but reduces the number of available samples. This cannot be

compensated by an augmentation of the observation time, which is limited by the assumption of stationary currents. Another challenge is the production of surface current maps obtained by resolving the received signals in range and azimuth. While range gating is always achieved with a standard FMCW chirping technique, the azimuthal discrimination of surface currents is a more delicate task. Extended linear antenna arrays (classically done with BF techniques) allow sweeping the different bearings in the radar field of view. With cross-loop compact antenna systems, the directions of arrivals are obtained

through high-resolution DF methods such as MUSIC algorithm. These are based on a covariance analysis of the individual Doppler spectra received on each antenna, an operation which requires processing a sufficient number of sample spectra. For this reason, compact systems usually necessitate a longer observation time than extended arrays to obtain reliable surface current maps. The latter have a more "peaky" aspect than those obtained with BF and do not suffer from angular smoothing. However, some wrong or missing allocations of the directions of arrivals can make them very lacunary and spotted with

many outliers. The quality of azimuthal processing with compact systems also relies on a careful calibration of the complex antenna gains, a procedure that usually necessitates extra hardware deployment. A last factor that impacts on data quality is the frequent occurrence of Radio-Frequency Interferences (RFI) from external electromagnetic sources. The RFI produces



sharp artificial Doppler peaks in the direction of the source, which can be erroneously interpreted as Bragg peaks and lead to a strip of false values in the radial current map.

All in one, there are many factors that affect the "voltage to radial current" transformation and might degrade the quality of the resulting surface current maps. This often results in a poor spatial coverage due to lacunary estimated and limited SNR, outliers due to wrong allocations of direction of arrivals or RFI and smoothed, underestimated currents due to an insufficient angular (BF) or temporal (DF) resolution.

To mitigate these deficiencies the HFR currents generally undergo some a posteriori processing and quality checks as

described in Mantovani et al. (2020). However, very often this cannot fully compensate for the insufficient quality and coverage of data and can even produce realistic looking but incorrect artificial maps. It is therefore important to correct as much as possible the shortcomings of HFR currents at the early stage of the 'voltage to current' transformation in order to optimize a posteriori processing and minimize its artifacts.

In the last few years some promising ideas and techniques have been proposed to improve the quality of the raw HFR signal

processing. This includes new calibration techniques (e.g. Flores-Vidal et al., 2013), original antenna processing methods (e.g. Dumas and Guérin 2020, Guérin et al., 2021), use of bi-static and multi-static configurations (e.g. Dumas et al., 2020), efficient RFI rejectors (e.g. Tian et al., 2017; Gurgel et al., 2007), non-spectral estimators (e.g. Domps et al., 2020 and 2021). Fig. 3 shows an example of the amelioration that can be obtained with a non-standard array processing method based on antenna grouping and direction finding (Dumas and Guérin, 2020) over a classical beam forming in the case of the 12-

antenna receiving array of Fort Peyras (Toulon, southeastern France). As seen in Fig. 3, fine contrasted patterns of radial current are unveiled when resorting to such a high-resolution technique while maintaining a good spatial coverage.

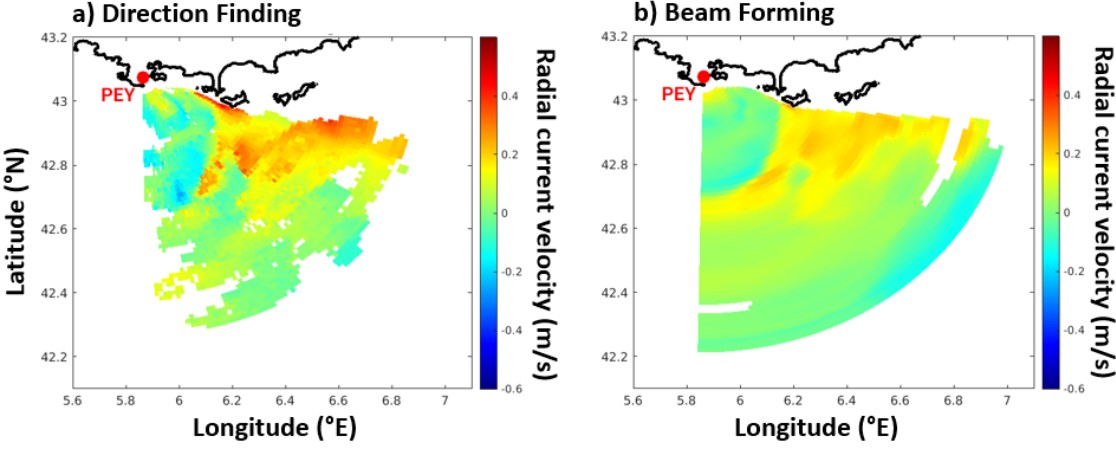

**Figure 3. Hourly radial surface current maps obtained on September 1st, 2020, 06:00 TU, with the HFR station deployed in Fort**
**Peyras (PEY) in Toulon (SE France) with a 12-antenna receiving array operating at 16.15 MHz. The range resolution is 1.5 km and the maximum range is about 80 km. a) Radial map obtained by using antenna grouping and self-calibration: fine and contrasted structures are unveiled; b) Radial map obtained using the classical beam forming azimuthal processing: small patterns are smoothed and contrast is reduced.**





The credibility of HFR-derived current data has been extensively proved in numerous coastal areas of the Mediterranean Sea by adopting Eulerian or Lagrangian approaches. Previous investigations included direct comparisons against independent in situ sensors like PCMs, moored ADCP's, drifters, ship-based sensors or similar (Cosoli et al., 2010; Berta et al., 2014; Lorente et al., 2014, 2015 and 2021; Corgnati et al., 2019a; Lana et al., 2016; Kalampokis et al., 2016; Capodici et al., 2019; Guérin et al., 2021, Molcard et al., 2009; Bellomo et al., 2015).

When the HFR footprint overlooks a moored instrument within its spatial coverage (Fig. 4, a), an accuracy assessment of HFR surface currents can be performed with radial or total vectors. In the first case, the HFR radial arc geographically closest to the in situ instrument location is selected for each HFR site and radial current vectors estimated at each arc point have been compared with the radial projection of PCM velocities (Cosoli et al., 2010; Lorente et al., 2014 and 2015). This comparative analysis allows the computation of statistical parameters (e.g., the correlation -CORR- and the root mean

squared error -RMSE-) as a function of the angle between the buoy and the arc grid point position. In the absence of direction-finding errors (DF), maximum CORR and minimum RMSE values should be found over the arc point closest to the buoy location. In the presence of DF, the bearing offset is thus expressed as the angular difference between the arc point with maximum correlation and the buoy location (Fig. 4, b). In the second case, HFR total vector hourly estimations at the grid point closest to the buoy location are compared against in situ current datas to provide upper bounds on the HFR accuracy.

Comparisons are commonly undertaken using zonal (U) and meridional (V) components in order to evaluate the agreement between both instruments (Fig. 4, c).





**Figure 4. Validation of hourly surface currents provided by the HFR deployed in Ebro Delta (NW Mediterranean Sea), shown in**
**Fig. 1, against a buoy for May-October 2014 (Lorente et al., 2015). a) Example of hourly map of surface current circulation. Pink**
**dot and purple squares represent the buoy and HFR sites location, respectively; b) Validation of radial currents. Correlation (solid**
**line) and RMSE (dashed line) between radial currents estimated by the buoy and those measured by SALO HFR site. The vertical**
**dotted line represents the angular position of the moored buoy. The vertical red solid line denotes the angular position of**
**maximum correlation (CORR), which is gathered with the associated RMSE and bearing offset (in red) values; c) Low-pass**
**filtered time series of zonal and meridional currents measured by the buoy (blue line) and HFR at the closest grid point (red line).**



Supplementary validation works with radial measurements were carried out in the Mediterranean Sea when the geometry of the emplacement gave the chance to perform a self-consistency analysis on the radar-to-radar overwater baselines in order to evaluate intrinsic velocity uncertainties in HFR radial velocities (Lorente et al., 2014). This methodology, previously applied

in other parts of the world (Paduan et al., 2006; Atwater and Heron, 2010; Gómez et al., 2020), states that in the absence of errors two facing HFR sites should provide the same estimates of radial velocities (differing only in the sign) at the midpoint of the baseline that joins them, since the range and the angular distribution are similar. This self-consistency test presents some benefits like the nonexistence of horizontal scale or depth mismatch, as the two involved sites are operating in the same frequency, providing two currents datasets with, in principle, identical origin and nature.

In terms of lagrangian assessment, it is worth mentioning that the Tracking Oil Spill and Coastal Awareness (TOSCA) project experience (Bellomo et al., 2015) constituted one of the first coordinated initiatives at Mediterranean level to test the precision of a core of 12 HFRs and identify a set of good practices for pollution mitigation. Among other valuable goals, the 5-country TOSCA experiment aimed at comparing HFR-derived measurements against the trajectories provided by 20 CODE-type drifters (Davis, 1985), which were drogued in the first upper meter of the oceanic layer and acted as proxies for

substances passively advected by currents. In all cases, the RMSE of the radial velocity difference between HFR and drifters lied in the range of 5–10 cm·s$^{-1}$, which are in line with previous literature given the expected variability at the HFR subgrid level.

As an overall summary of the validation works, RMSE and CORR values have been typically reported to emerge in the ranges 5-20 cm·s$^{-1}$ and 0.32-0.92, respectively. Relative HFR velocity errors can vary widely depending on the

characteristics of the site, the radar transmission frequency, the sensor type and location within the sampled domain, as well as the data processing scheme used (Rypina et al., 2014; Kirincich et al., 2012).

These validation studies acknowledged that observed discrepancies between HFR-in situ estimations might be partially attributable to the combined contribution of several factors such as the mismatch in time sampling and averaging, distinct horizontal averaging scales, contributions from Stokes drift likely included in HFR-derived estimates or the influence of

Ekman stratification in the current profile, subsequently leading to potential velocity differences in the upper water column (Laws et al., 2003; Ohlmann et al., 2007; Chapman et al., 1997; Kohut et al., 2006). In this frame, the instrumental noise and sub-grid scale current variability have been also documented to yield noise levels of 4-6 cm·s$^{-1}$ (Emery et al., 2004; Ohlmann et al., 2007; De Paolo et al., 2015).

**2.2.2 Wave measurement retrieval from HFRs**

In addition to surface ocean currents, HFR directional wave spectrum and derived parameters such as local significant wave height, centroid wave period and mean wave direction can be determined from the weaker second-order sea-echo Doppler spectrum by adopting two main approaches: full integral inversion or fitting with a model of ocean wave spectrum (Lipa and



Nyden, 2005). A variety of inverse techniques have been developed over the last decades (Barrick, 1977; Wyatt, 1990; Hisaki, 2006).

Wave measurements derived from HFR have a broad range of potential applications and can be used as input data for numerical models' validation (Saviano et al., 2020a), assimilation into wave models (Siddons, Wyatt and Wolf, 2009; Waters et al., 2013), for wave energy harvesting (Ramos, Graber and Haus, 2009) or the analysis of extreme wave height events (Lorente et al., 2021). HFR wave data can provide assistance to maritime navigation and wise decision-making, from both commercial and recreational aspects, by identifying severe sea states in densely operated maritime areas where fixed in

situ moorings may be compromised (e.g., at the entrance of congested harbours, first-order spots in terms of activity and trade volume). Furthermore, HFRs can help detect the interaction between high incoming waves, intense river outflow currents and wind-forced flow over the inner continental shelf, as highlighted by Lorente et al. (2021).

In order to infer how much confidence can be placed in wave parameters retrieved by HFR systems, their accuracy must be evaluated under different sea states and coastal configurations (Fig. 5). Previous validation experiments, some of them listed

in Table 1, included comparisons against independent in situ observations, remote-sensed wave estimations or numerical outcomes over a variety of regions in the Mediterranean Sea such as the GoN (Falco et al., 2016; Saviano et al., 2019, 2020a and 2020b), Sicily (Orasi et al., 2018) or the Ebro Delta (Lorente et al., 2021). Regardless of the manufacturer, the operational frequency and the methodology used to determine wave parameters, the positive contribution of commercial HFR systems to characterize the main wave patterns (and the related spatio-temporal variability) has been unequivocally

proven under both standard metocean conditions and severe sea states.

| Reference | Orasi et al. (2018) | Saviano et al. (2019) | Lorente et al. (2021) |
|---|---|---|---|
| Frequency (MHz) | 13.5 | 25 | 13.5 |
| Study Area | Malta-Sicily Channel | GoN | Ebro River Delta |
| Time Period | Winter 2016-2017 | Entire 2010 | 19-24 of January 2020 |
| Validation against | satellite altimeter | Buoy | buoy |
| Parameter | SWH | SWH | SWH, TM and direction |
| Metric for SHW | CORR = [0.86-0.98] MSE = [0.04-0.29] | CORR = [0.50-0.75] RMSE = [0.20-0.66] | $RMSE_N$ = 0.12 Skill Score = 0.93 |

**Table 1. Review of the most recent studies about validation of High-Frequency radar (HFR) derived wave parameters against independent wave observations. Skill metrics obtained for the significant wave height (SWH) during the studied period included the mean squared error (MSE), the root mean square error (RMSE), the normalized RMSE ($RMSE_N$), the Pearson's correlation**





**435** **coefficient (CORR) or the Skill Score (SS) proposed by Wilmott (1981). The metrics intervals denote the range of results obtained for several sites composing each HFR system.**

A widely accepted approach with DF systems consists of comparing HFR wave estimations, extracted along several annular rings (circular concentric range arcs) centered in each of the HFR sites, against in situ observations to quantify the degree of

**440** accuracy as a function of the distance to the shoreline. As shown by Saviano et al. (2019) and Lorente et al. (2021), wave estimations are often averaged among the intermediate range arcs to improve data quality and availability. This constitutes an optimal operational trade-off, as it guarantees the highest number of recordings. While close enough to the shoreline (so as the sea echo intensity is sufficiently high to ensure good data quality), the range arcs are also deep enough to avoid shallow water effects on radar sea-echo: wave breaking and the decrease the saturation limit on wave height as ocean depth

**445** decreases (Lipa et al., 2008). In the case of linear phased-array BF systems, consisting of at least 12 antennas, they can provide maps of wave parameters with the same spatio-temporal resolution as with surface currents (Gómez et al., 2015).

According to the skill metrics presented in Table 1, which are in accordance with previous validation exercises reported in other European waters (Basañez et al., 2019; López and Conley, 2019; López, Conley and Greaves, 2016; Gómez et al., 2015; Long et al., 2011), it can be concluded that properly treated HFR-derived wave estimations can be potentially

**450** employed for operational coastal monitoring across a wide range of sea states. Ad hoc quality control methodologies, based on the particular local environment, are required to ensure robust HFR wave measurements. Although the precision and availability of HFR-derived wave estimations have been documented to be lower during calm sea states (as the second-order spectrum is closer to the noise floor), HFR might act as an effective coastal monitoring assets, especially in locations where in situ devices cannot be deployed (such as harbour entrances) or when in situ wave observations are temporarily unavailable

**455** due to instrument outages or breakdowns.

Particularly for the Mediterranean coastal waters, the performance of the HFR system installed in the Gulf of Naples (GoN hereinafter) in the retrieval of wave parameters has been tested in different works, aimed at providing on the one hand an assessment of the accuracy of HFR measurements, and on the other a reconstruction of the wave climatology of the basin. The validation of HFR wave data has been accomplished using two different platforms, namely an *in situ* buoy and two

**460** wave models. In the first case (Saviano et al., 2019), recordings from an offshore directional buoy (located outside the areal coverage of the HFR system) were used to evaluate the agreement with the patterns depicted by the three HFR sites building the GoN network (Fig. 1), over a one-year reference period. As reflected in Fig. 5a, the comparison indicated that both platforms returned collimating descriptions of the wave field under both calm and stormy periods, and that the HFRs could retrieve realistic measurements also above the theoretical maximum recordable wave height (Lipa and Nyden, 2005).

**465** Additional insight into the validity of HFR data has been gained by the comparison agaings wave measurements provided by two models, WAVEWATCH III and SWAN (Simulating WAves Nearshore), over a three-year period (Saviano et al., 2020a). Overall, the HFR and model data were consistent, although discrepancies in lower sea states and in extreme conditions could be reported. The validation of HFR measurements was a fundamental prerequisite to extract long-term





information on the characteristics of the wave field in the GoN and exploit them to build a wave climatology of the basin. To

this aim, wave measurements from the HFR network over four-and-half years were complemented with records from an ADCP interlocked with a Monit-Med (MEDA) elastic beacon collected over almost three years (Saviano et al., 2020b).

The integration of the results gathered through these works allow depicting some peculiarities of the wave field in the GoN, namely: (i) a predominantly wind driven wave field, with specific seasonal recurrent patterns; (ii) the occurrence of more energetic conditions in autumn/winter, particularly in association with low-pressure systems acting over the region; (iii) the

establishment of a stable calm state, driven by spring/summer breeze regime; (iv) the directional distribution of approaching waves depending on the sub-basin of the GoN considered, corresponding to the different sectors covered by each HFR site. These patterns are comparable over the years (Falco et al., 2016; Saviano et al., 2019, 2020a and 2020b), but at the same time are coherent with the typical climate of the Southern Tyrrhenian Sea and with previous studies carried out in the GoN. In addition to insights strictly focusing on the basin dynamics, the outcomes collected in the GoN demonstrates that the

HFRs provide reliable measurements of waves, particularly in terms of significant wave height. With reference to wave period, DF system returns a centroid period which falls between the mean and peak periods typically retrieved by other platforms (Saviano et al., 2019 and 2020b). As such, the centroid period can be used as a robust estimator in line with what discussed in Long et al. (2011). In a more general framework, the positive experience matured in the GoN demonstrates that HFRs should be considered an integral part in the design and implementation of coastal monitoring systems thanks to their

ability of reconstructing not only the surface current field, but also wave dynamics and wind. The performance of HFR systems, however, still needs to be improved as discussed in Saviano et al. (2019), for example by standardising QA/QC protocols, optimising inversion methods and wave retrieval algorithms.

In the Malta-Sicily Channel, Orasi et al. (2018) compared significant wave height measurements from 4 HFR sites against both numerical simulations (provided by 1/60º Mc-WAF system, based on WAM model) and satellite altimeter data (i.e.

Jason2, Jason3 and SAR Saral Altika missions). As shown in Fig. 5b, better agreement is achieved in the intermediate rings with respect to the HFR site location and particularly when compared versus altimeter data. WAM slightly underestimates the SWH during a storm event occurring along the analyzed time series and with a return period of 4 years.

In terms of extreme events, record-breaking storm Gloria (January 19-24, 2020) hit the NW Mediterranean Sea with heavy rainfall, strong easterly winds and very high waves (Lorente et al., 2021). Although the low-lying Ebro Delta region (Fig. 4a)

was severely inundated, the HFR deployed there was able to effectively monitor Gloria´s striking features. As shown in Fig. 5c, the visual resemblance between in situ data and HFR-derived estimations of SWH (from ALFA site) is remarkably high. The peak, which was well captured in terms of intensity (7.28 m) and timing, fairly exceeded the percentile 99 derived from the buoy estimations for a 15-year period (2004-2019), established at 2.87 m.

For phased-array HFR, the reconstruction of the wave field from the backscattered signal can be attempted by using a single

station (Fig. 5d). Depending on the method used, this can provide different estimations of the wave frequency spectrum, from which integrated parameters can be estimated such as significant wave height and wave period. Nevertheless, both approaches inverting the nonlinear integral equation of radar cross-section and the more simplified empirical approaches



result in an ambiguity of the directional spectrum solution (Hisaki, 1996; Gurgel et al., 2006). Therefore, to solve this ambiguity and to be able to provide directional wave information, a second HFR site overlooking the same ocean patch from
a different direction is required.

An evaluation of wave parameters measured by a single HFR station concluded that significant wave height estimates are not robust when the waves propagate roughly perpendicular to the radar beam. In such cases, which did not present often, a different algorithm can be used which improves the estimations. Since there is no directional information provided by a single HFR, there is no way to select between the two algorithms solely by using the measured data. It was shown that dual
radar estimates are more accurate than using single radar site estimations (Wyatt, 2002).

In order to solve the above discrepancy, de Valk et al. (1999) took into account additional physics. Their reconstruction method inverts the Doppler backscatter integral together with a reconstruction of the wave field using the wave action equation while neglecting ambient currents and various source functions. Hisaki (2006) extended de Valk et al.'s approach to include also the wind input, dissipation and nonlinear interaction source terms. Both require solving an iterative and
location-taylored model. A more recent work (Alattabi et al., 2019) provides a model which treats swell and wind waves separately combining former works into a single empirical hybrid model. Its results using a single VHF station provided good correlation to various in-situ measurements. This application has some limitations for nearshore swells, but its accuracy and simplicity show good perspectives for a large-scale adaptation after confirmations using radar systems of different frequencies.





**Figure 5. (a)** Time series of SWH provided by HFR-GoN (SORR site) and buoy deployed in the GoN (Fig. 1). Timelines of data availability provided at the top. Dotted lines represent the theoretical upper and lower detectability thresholds of this HFR; **(b)** Time series of SWH (averaged in time and space) provided by HFR-Calypso (arc 3 of Barkat site), WAM model and altimeter during a storm in the Malta-Sicily Channel; **(c)** Time series of SWH provided by HFR-DeltaEbro (arcs 4-9 of ALFA site) and B1 buoy during storm Gloria in the NW Mediterranean Sea; **(d)** Time series of SWH (~20km off-shore Israel) as measured by HFR-Israel (Fig. 1). Preliminary uncalibrated data using Gurgel and Schlick (2006) were compared with ERA5 wave reanalysis (from the ECMWF center).

A portion of the detected discrepancies in wave measurements could be attributed to: i) the mathematical inversion process of the second-order is unstable and diverges rapidly from the true solution in presence of noisy data. ii) the assumptions made in the inversion method. The Pierson-Moskowitz fit-to-spectrum unimodal model used has previously proved its validity to properly describe wind-dominated seas and also swell dominated seas, whereas this might be different under some combination of multi-modal sea-states under complex met-ocean conditions; iii) the different sampling techniques. Whereas

DF HFR systems provide wave data averaged over range rings (assuming homogeneity over the whole of each circular range





cell), buoys give point measurements. In this context, coastal effects can also lead to locally varying wave fields and make absolute comparisons between in situ and remote-sensing instruments even harder.

### 2.2.3 Wind measurement retrieval from HFRs

While the analysis of surface currents and the retrieval of wave parameters are well established, the worldwide use of first-

order peaks to measure wind direction still remains less explored (Heron and Rose, 1986; Heron, 2002; Kirincich, 2016; Hisaki, 2017; Wyatt, 2018). Besides, additional efforts should be devoted in the short-term future to the development of robust algorithms for a reliable measurement of wind speed (Shen et al., 2012, Vesecki et al., 2002).

To the best of our knowledge, Saviano et al. (2021) constitutes the first attempt to analyze HFR-derived wind direction in the Mediterranean Sea, using a 25 MHz DF system. HFR measurements were compared with in situ recordings from a weather

station in the GoN, revealing: i) the potentialities of this remote-sensing technology as a monitoring platform when the wind speed exceeds a 5 m·s$^{-1}$ threshold; ii) the relevance of the operational frequency in the accuracy of HFR wind measurements, with higher frequencies leading to estimations that are in better agreement with in situ measurements, as previously indicated by Shen and Gurgel (2018). This is due to the fact that the resonant Bragg waves have a shorter wavelength and thus are more sensitive to changes in the wind direction field.

The first model to extract the wind direction from HFR backscatter was suggested by Long and Trizna (1972). In recent decades, different research groups developed algorithms for the extraction of wind direction (Zeng et al., 2018; Chu et al., 2018; Hisaki, 2017; Kirincich, 2016; Shen et al., 2012; Heron, 2002; Huang et al., 2004; Gurgel et al., 2006) and more recently also a neural network method was applied for wind field inversion (Zeng et al., 2016).

Although works and publications dealing with HFR wind measurements are still scarce compared to those analyzing HFR

currents or waves, several examples presenting and validating HFR wind direction data can be found in the literature (Heron, 2002; Lipa et al., 2014; Kirincich, 2016; Hisaki, 2017; Shen and Gurgel, 2018; Wyatt et al., 20061, Wyatt, 2018; Saviano, 2021). Usually, the experiments are carried out in a wide coastal area and over the ocean sea for periods from days to months.

Some of these previous studies affirm that the accuracy of HFR wind direction measurements is related to many factors

(Lipa et al, 2014). Diverse studies on the comparison with in situ measurements acknowledged that with wind speeds lower than 5 m·s$^{-1}$ the reversal of the wind direction and hence HFR derived wind direction is not reliable (Lipa et al., 2014, Wyatt, 2018; Shen and Gurgel, 2018). This is mainly due to the fact that at high wind speeds, the direction of the Bragg resonant waves (i.e. the HFR-derived wind direction) agrees better with the wind direction (Shen and Gurgel, 2018). Another important factor is the frequency of the HFR, since HFR systems operating at higher frequencies leads to wind direction

measurements that are in better agreement with in situ ones. This is due to the fact that the corresponding resonant waves (i.e. half the radio wavelength) are relatively shorter ones being more sensitive to a change in wind direction, rapidly responding to local wind excitation and variability (Shen and Gurgel, 2018). In addition, an accurate knowledge of the seasonal wind field of the study area is fundamental to assess the correct investigation.



In the Ligurian Sea experiment a pattern-fitting method for wind direction inversion from a 12 MHz beam forming HFR was presented in Shen and Gurgel (2018). A meteorological buoy provides the in situ wind speed data from 10 May 2009 to 8 June 2009. During the experiment the wind speed was relatively low, only 18.9% of wind records exceeded 5 m·s⁻¹. Results show that, for wind direction measurements from HFR backscatter, the accuracy strongly depends on the radar frequency and from the measurement of wind speed using buoys, under higher-wind conditions, the inversion of wind direction is better.

The analysis in GoN (southern Tyrrhenian Sea), in an intricate coastal area with very special local factors influencing the wind field show comparisons between HF wind direction, in situ measurement (weather station) and model SKIRON/Eta in selected events (Saviano et al., 2020a and 2020b). As shown in Fig. 6, the comparisons reveal a good statistical agreement between the platforms with robust values of circular correlation coefficient, during winter events where the wind speed exceeded the threshold of 5 m·s⁻¹ for a period of 72 hours (for circular statistics applied to HFR data see Ranalli et al., 2018). Furthermore, the acquisitions of all range cells (RCs) or annulus around the HFR sites were investigated: in all the events, the RCs near the coast and the offshore ones give poorer statistical results compared to the central RCs, while the best agreement is found between 4 and 10 km from the coast (Fig. 6).

From this investigation we can draw several conclusions: i) the inversion of wind direction is in general not reliable at low wind speeds; ii) additional investigations on noise interference in the returned signal with the inversion method of wind direction are still necessary and other important physical effects on the radar inversion should be evaluated, such as wind duration and fetch.



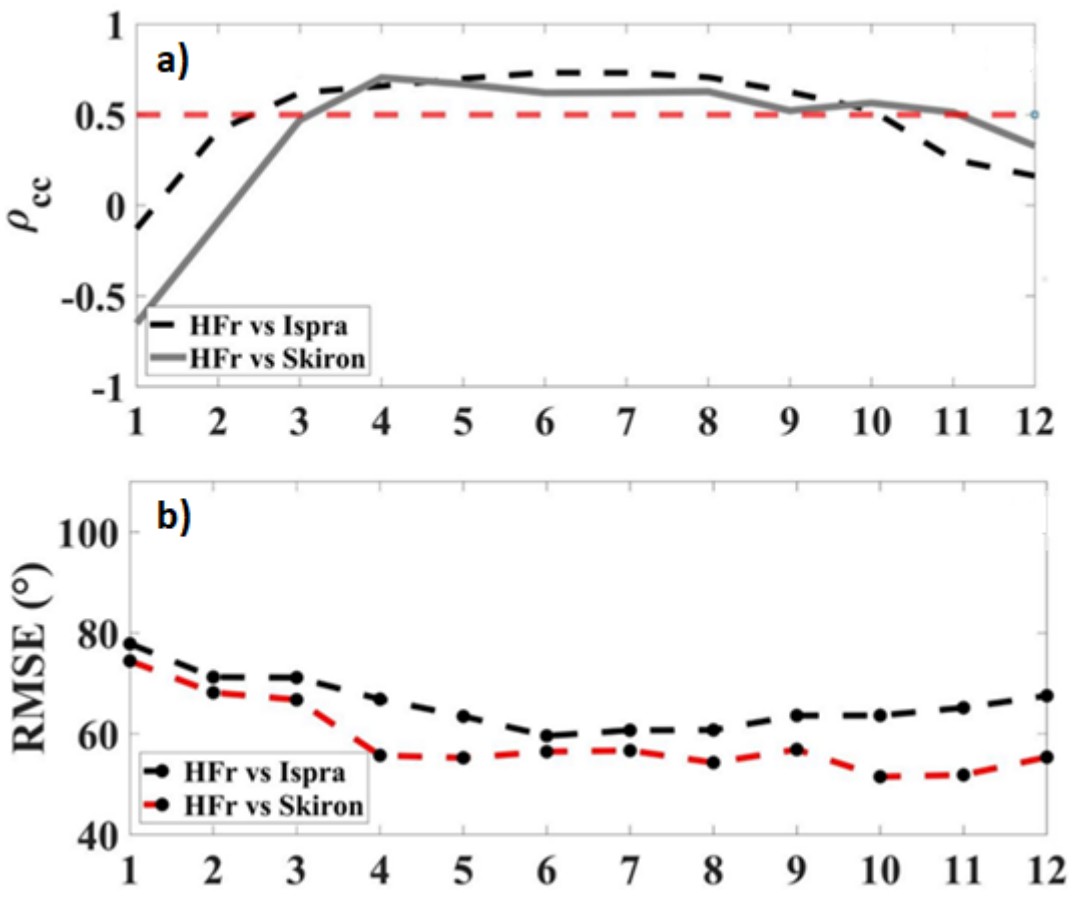

**Figure 6. Variation of a) circular correlation coefficient (ρcc) and b) RMSE on different range cells (km from site), derived from the comparison of HFR-GoN wind direction against data provided by a weather station (Ispra) and a numerical model (Skiron) for the event occurred during 6-8 February 2009 in the GoN.**

## 2.3 Best practices

A key element for an effective exploitation of a large-scale HFR network, especially when operated by many different players, is the implementation of common guidelines and best practices recognized by the international community. This level of harmonization ensures that all the sites are deployed and operated with a similar and sufficient standard of quality and thereby allows researchers to assess the consistency of results when performing data analyses and applying new methodologies on different sites and geographical areas. The availability of such manuals also improves the sustainability of the HFR network, since they facilitate the exchange of know-how between partners and help new actors to integrate their systems with minor effort.



Harmonization of HFR systems management is also a requirement for delivering robust operational products and services. An effort has been done in Europe for reviewing and complementing existing best practices related to surface currents retrieval (Mantovani et al., 2020). The Mediterranean HFR community has been actively involved in this task, especially in the framework of the EuroGOOS HFR Task Team and the H2020 project JERICO-Next, and will benefit from the

progressive implementation of the defined recommendations.

The guidelines are shaped considering the general HFR principles of operation, independently from the commercial manufacturer or antenna design and setup, and they include:

i)   Site requirements for optimal HFR performance

ii)  Typical authorizations needed for installing and operating an HFR station

iii) List of basic accessories for ensuring protection of the equipment, remote management and reliable data transfer

iv)  Items to be evaluated for estimation of deployment and operating costs

v)   Key elements for a correct setup of HFR systems and suggestions for monitoring their performances

vi)  Scalable data management, encompassing a common protocol for data processing and the standardization of a single HFR interoperable data format with a unified list of metadata descriptors.

vii) Unified procedures for quality assurance - quality control (QA-QC) of HFR data in real-time.

**2.4 Data flow: from providers to distribution via the EU HFR Node**

In 2014, EuroGOOS launched the HFR Task Team (Fig. 2) with the aim of promoting the HFR technology in Europe. This was the cornerstone of a fruitful and still ongoing roadmap towards the unlocking of the unprecedented potential of HFRs for an integrated coastal management and its application to a wide range of maritime sectors, such as SAR, renewable energy,

fishery management, tracking of pollutants, or improvement of ocean forecast models through data assimilation schemes.

Indeed, this first step followed up on many initiatives in Europe aiming at building an operational HFR European network based on coordinated data management for the development of operational ocean monitoring via HFR systems, and integration of HFR products into the major platforms for marine data distribution. In 2015, a pilot action coordinated by EMODnet Physics began to develop a strategy for assembling HFR metadata and data products within Europe in a uniform

way to make them easily accessible, and more interoperable (Fig. 2). The EU project JERICO-NEXT, launched in 2015, aimed to provide procedures and methodologies to enable HFR data to comply with the international standards regarding their quality and metadata, within the overall goal of integrating the European coastal observatories. In parallel, the SeaDataCloud EU project, launched in 2016, contributed to the integration and long-term preservation of historical time series from HFR into the SeaDataNet infrastructure (Fig. 2) by defining standard interoperable data and Common Data Index

(CDI) derived metadata formats and Quality Control (QC) standard procedures for historical data. In 2016 as well, the CMEMS Service Evolution Call supported the INCREASE project, which set the bases for the integration of existing European HFR operational systems into the CMEMS-INSTAC. More recently, the EU projects Jerico-S3 and EuroSea are





continuing these efforts for further expanding the standardization and interoperability of HFR data in order to promote the distribution of high quality HFR data and improve their impact in scientific, operational and societal applications.

The results of these integrated efforts are significant and allowed the achievement of the harmonization of system requirements and design, data quality and standardization of HFR data access and tools (Mantovani et al., 2020). The European standard format for HFR data and metadata model has been defined and implemented, compliant with Climate and Forecast Metadata Convention version 1.6 (CF-1.6), OceanSITES convention, CMEMS-INSTAC and SDC requirements and INSPIRE directive. Furthermore, a battery of the QC tests to be mandatorily applied to HFR data has been defined

according to the EuroGOOS Data Management, Exchange and Quality Work Group (DATAMEQ) working recommendations on real-time QC and building on the Quality Assurance/Quality Control of Real-Time Oceanographic Data (QARTOD) manual produced by the US Integrated Ocean Observing System (IOOS). Thanks to these achievements, the inclusion of HFR data into CMEMS-INSTAC, EMODnet Physics and SDC Data Access was decided to ensure the improved management of several related key issues as Marine Safety, Marine Resources, Coastal and Marine Environment, Weather,

Climate and Seasonal Forecast.

The EU HFR Node (Fig. 7) was established in 2018 by AZTI, CNR-ISMAR and SOCIB, under the coordination of the EuroGOOS HFR Task Team, as the focal point and operational asset in Europe for HFR data management and dissemination by promoting networking between EU infrastructures, marine data portals and the Global HFR network. The EU HFR Node is fully operational since December 2018 to distribute tools and support for standardization to the HFR providers as well as

standardized Near Real Time (NRT) and delayed-mode HFR radial and total current data to CMEMS-INSTAC, EMODnet Physics and SDC Data Access. Within the European framework, the EU HFR Node is now managing data from 12 HFR networks (built by 35 radar sites) and is expected to manage 17 networks (for a total of 50 radar sites) by mid-2021. In particular, 17 of these 50 sites (34%) are deployed in the Mediterranean coastline and belong to the MONGOOS network: HFR-Gibraltar, HFR-Ibiza, HFR-DeltaEbro, HFR-TirLig and HFR-NAdr (Fig. 1). Furthermore, the EU HFR Node

integrates and delivers US HFR network data to the aforementioned data portals. In particular, the EU HFR Node implements the operational chain which encompasses data acquisition and harvesting, harmonization, formatting, QC, validation/assessment, NRT data delivery and historical data distribution with different reprocessing levels.

The core of this service consists in the continuous development of the data model and the processing standards through discussion with operators, providers, distributors and international experts. Based on this, the EU HFR Node maintains and

updates manuals, procedure guidelines and software tools, and pushes them towards the HFR operators, providers and managers via repositories and training workshops. In particular, the software tools for processing native HFR data for QC and converting them to the standard format for distribution are continuously made available to HFR operators via public GitHub repositories and releases with DOI assigned (DOI:10.5281/zenodo.2639555).

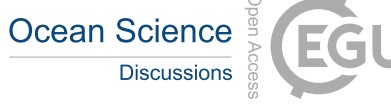

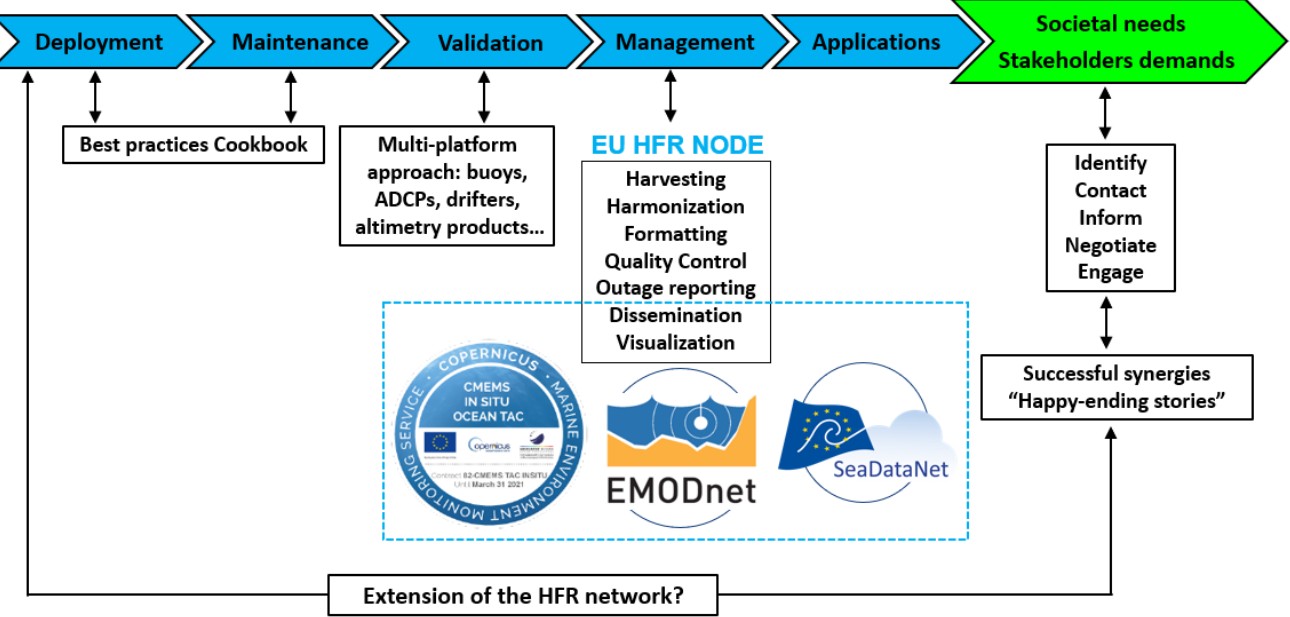


**Figure 7. Basic roadmap for the homogenization and distribution of HFR data, from the data providers to the end-users.**

The data processing and distribution service is founded on a simple and very effective rule: if the data provider can set up the data flow according to the defined standards, the node only checks and distributes the datasets. If the data center cannot set

up the data flow, the EU HFR Node directly harvests the raw data from the provider, harmonizes, quality-controls and formats these data and makes them available to the marine data portals. The strength and flexibility of this solution reside in the architecture of the European HFR node, which is based on a centralized database, fed and updated by the operators via a webform (http://150.145.136.36). The database contains updated metadata of the HFR networks and the needed information for processing/archiving the data. Finally, the guidelines on how to set the data flow from HFR providers to the EU HFR

Node are thoroughly described in Reyes et al. (2019).

**2.5 The European common data and metadata model for real-time High Frequency Radar surface current data**

An appropriate file description (i.e. "comprehensive metadata"), complying with accepted standards, is crucial for enforcing data discovery and access. The detailed metadata description is a prerequisite for the fully operational implementation, providing an inventory of the continuously available data for operational models. It is also necessary for providing a detailed

overview of marine monitoring programmes relevant for the Marine Strategy Framework Directive (MSFD) implementation. In the framework of the aforementioned initiatives and projects, in particular within Jerico-Next and INCREASE projects, a model for HFR derived data and metadata was defined and later implemented to be the official European standard for HFR





real-time data in order to ensure efficient and automated HFR data discovery and interoperability with tools and services across distributed and heterogeneous earth science data systems.

The model has been implemented according to the standards of Open Geospatial Consortium (OGC) for access and delivery of geospatial data, and is compliant with: i) the Climate and Forecast Metadata Convention CF-1.6; ii) the Unidata NetCDF Attribute Convention for Data Discovery (ACDD); iii) the OceanSITES convention and iv) the INSPIRE directive. The definition of the European common data and metadata model for real-time HFR data follows the guidelines of the DATAMEQ working group and fulfils the recommendations given by the Radiowave Operators Working Group (US
ROWG).

The model specifies the file format, the global attribute scheme, the dimensions, the coordinates, the data and QC variables and their syntax, the QC procedures and the flagging policy. The file format is the netCDF-4 classic model with the recommended implementation based on the community-supported CF-1.6.

Global attributes from Unidata's NetCDF Attribute Convention for Data Discovery (ACDD) are implemented and they are
divided in three categories: i) Mandatory Attributes for compliance with CF-1.6 and OceanSITES conventions; ii) Recommended Attributes for compliance with INSPIRE directive and iii) Suggested Attributes, that can be relevant in describing the data. Attributes have to be also organized by function: Discovery and Identification, Geo-spatial-temporal, Conventions used, Publication information and Provenance.

Variables are divided in three categories: i) Coordinate Variables orienting the data in time and space (they may be
dimension variables or auxiliary coordinate); ii) Data Variables containing the actual measurements and information about how they were obtained and iii) QC variables containing the Quality Control flag values resulting from the QC tests performed on the data. Variable short names from SeaDataNet (SDN) P09 controlled vocabulary are recommended. CF-1.6 standard_names are required, when available. The European common data and metadata model for real-time HFR surface current data is comprehensively described in the Jerico-Next Deliverable D5.14 (Corgnati et al., 2018).

In order to fulfil the specific requirements of CMEMS-INSTAC, EMODnet Physics and SDC Data Access, that are operationally distributing NRT and historical HFR data since 2019, the standard data and metadata model was declined for those specific applications: the manual for the standard data and metadata model adopted in CMEMS-INSTAC and EMODnet Physics is described in (Carval et al., 2020), the one for SDC Data Access is described in (Corgnati et al., 2019b).

## 2.6 Quality Control procedures

The European common data and metadata model for real-time HFR data requires a battery of QC tests in order to ensure the delivery of high quality data and to describe in a quantitative way the accuracy of the physical information and to detect occasional non-realistic current vectors or artefacts (defined as spikes, spurious values or unreliable data), generally detected at the outer edges of the HFR domain and flagged in accordance with a pre-defined protocol. These mandatory QC tests, based on DATAMEQ working recommendations on real-time QC and on the QARTOD manual, have been selected in strict



collaboration with most of the European HFR operators and data providers. While they are meant as a minimum set of QC needed for data distribution, any further QC processing of HFR data is strongly encouraged.

These standard sets of tests, which are manufacturer-independent, have been defined both for radial and total velocity data. The battery of mandatory QC tests and the flagging scheme are thoroughly described in Corgnati et al. (2018). Each QC test results in a flag related to each data vector: the flag is contained in the specific test variable. These variables can be matrices

with the same dimensions of the evaluated data variable, containing, for each cell, the flag related to the vector lying in that cell, in case the QC test evaluates each cell of the gridded data, or a scalar, in case the QC test assesses an overall property of the data file. An overall QC variable reports the quality flags related to the results of all the QC tests: it is categorized as a "good data" flag if and only if all QC tests are successfully passed by the data.

The mandatory QC tests for HFR radial velocity data are: Syntax, Over water, Variance threshold, Velocity threshold,

Median filter, Temporal derivative, Average Radial Bearing and Radial count.

The mandatory QC tests for HFR total velocity data are: Data density threshold, GDOP threshold, Variance threshold, Velocity threshold and Temporal derivative.

However, the main drawback lies with the potential removal of accurate data when the discriminating algorithm is based on tight thresholds. Therefore, HFR operators will need to select, and keep updated, the most suitable thresholds for some of

these tests. Since a successful QC effort is highly dependent upon selection of the proper thresholds, this choice cannot be done arbitrarily. Some fine-tuning, based on the specific historical conditions of the system, is thus required to have the right trade-off between confirmed outlier identification and false alarm rate, maximizing the benefit of the applications of these methods.

## 3. HFR systems in the Mediterranean Sea

The Mediterranean HFR network includes 15 different systems, which cover a small portion of the entire coastal domain (Fig. 1). Basic technical aspects of these systems are gathered in Table 2. The monitoring capabilities appear to be spatially asymmetric, with the concentration of HFR installations generally decreasing from NW to SE due to a wealth of political and socio-economic factors. Diverse interlinked aspects influenced in the selection of the place to deploy such HFR systems, namely: i) gaining access to suitable and unobtrusive emplacements, where electromagnetic interferences (from the

surrounding environment or the nearby presence of metal items, buildings or orographic obstacles) are inexistent or, at least, minimized; ii) in the case of academia, the proximity to the research laboratory in charge of the maintenance and scientific exploitation of such HFR system (which aids to mitigate the costs of prompt recovery in case of temporal outage); iii) the oceanographic interest of the selected coastal area (i.e., marine protected areas, biodiversity hotspots, etc.), where ocean processes of paramount importance take place at multiple spatiotemporal scales; iv) the societal concern tied to the HFR

location. Served as example, the Strait of Gibraltar (Fig. 1) constitutes a target for potential oil spill accidents due to both the



extremely intense maritime traffic (as the only entrance gate to the Mediterranean Sea from the Atlantic Ocean) and the significant trade volume related to the activity of the Port of the Bay of Algeciras.

| Name | Frequency (MHz) | Institution (Country) | Region |
|---|---|---|---|
| HFR-NAdr | 24.53 | OGS (Italy) and NIB (Slovenia) | Northern Adriatic Sea and the Gulf of Trieste |
| HFR-GoN | 25 | University Parthenope of Naples (Italy) | GoN in the Tyrrhenian Sea |
| HFR-LaMMA | 13.5 | Consorzio LaMMA (Italy) | Tuscany Archipelago in the Tyrrhenian Sea |
| HFR-TirLig | 26.28 and 13.5 | CNR-ISMAR (Italy) | Northern Tyrrhenian Sea and the Ligurian Sea |
| HFR-DeltaEbro | 13.5 | PdE (Spain) | Ebro river delta |
| HFR-Gibraltar | 27 | PdE (Spain) | Strait of Gibraltar |
| HFR-Ibiza | 13.5 | SOCIB (Spain) | Ibiza Channel |
| HFR-Calypso | 13.5 | Universities of Palermo (Italy) and Malta (Malta) | Malta-Sicily Channel |
| HFR-Calypso-South | 13.5 | University of Malta (Malta) | South of Malta island |
| HFR-SIC | 13.5 | OGS-CNR (Italy) | SW of Sicily island |
| HFR-MedTln | 16.15 | MIO and University of Toulon (France) | Western Ligurian Sea |
| HFR-MedNice | 13.5 | MIO and University of Toulon (France) | Western Ligurian Sea |
| HFR-Split | 26.28 | Institute of Oceanography and Fisheries (Croatia) | Middle Adriatic Sea |
| HFR-Israel | 8.30 | Tel-Aviv University (Israel) | SE Mediterranean Sea |
| HFR-Dardanos | 16.10 | University of the Aegean and Hellenic Centre for Marine Research (Greece) | Eastern coast of Lemnos Island |

**Table 2. Description of the HFR systems deployed in the Mediterranean Sea, which are currently working on an operational way.**
**The acronyms PdE, SOCIB, CNR-ISMAR, OGS, MIO and NIB stand for "Puertos del Estado", "Balearic Islands Coastal Ocean Observing and Forecasting System", "Institute of Marine Sciences of the National Research Council of Italy", "Istituto Nazionale**





di Oceanografia e di Geofisica Sperimentale", "Mediterranean Institute of Oceanography" and "National Institute of Biology", respectively.

In terms of current status, the Mediterranean HFR network is characterized by the presence of a considerable number of existing sites (47), 31 of them working operationally and 16 sites out of order permanently due to a variety of reasons ranging from technical to financial issues. In the short-term future, 13 new sites will be incorporated (Fig. 8, a). Broadly speaking, up to 82% of the deployments have been permanent, while a small portion of them were temporarily implemented in the frame of specific time-delimited research projects (Fig. 8, b). Finally, DF HFR are more abundant than BF systems in

this regional domain (Fig. 8, c).

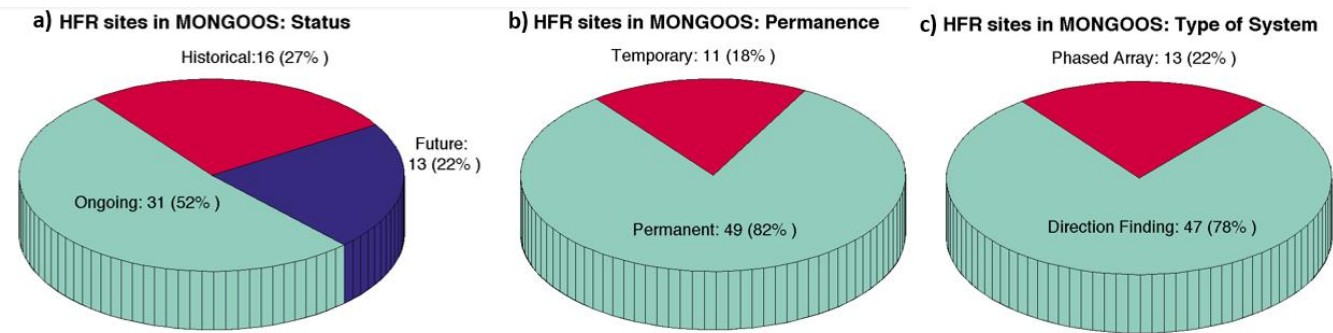

**Figure 8. Pie charts showing the number and percentages of HFR systems in MONGOOS in terms of status, permanence and type**

In comparison with other regional alliances like the Iberia-Biscay-Ireland Regional Ocean Observing System (IBIROOS), MONGOOS fairly represents the 55% of the total HFR sites in Europe (Fig. 9, a). Several of those MONGOOS networks (about the 23%) are already integrated in the EU HFR Node data flow, thus providing standardized and interoperable near real time (NRT) datasets to the CMEMS-INSTAC, EMODnet Physics and SDC distribution platforms (Fig. 9, b). However, a smaller fraction of them (15%) are already delivering reprocessed (REP) data (Fig. 9, c). Notwithstanding, new

connections are foreseen to the EU HFR Node in the incoming months of 2022.





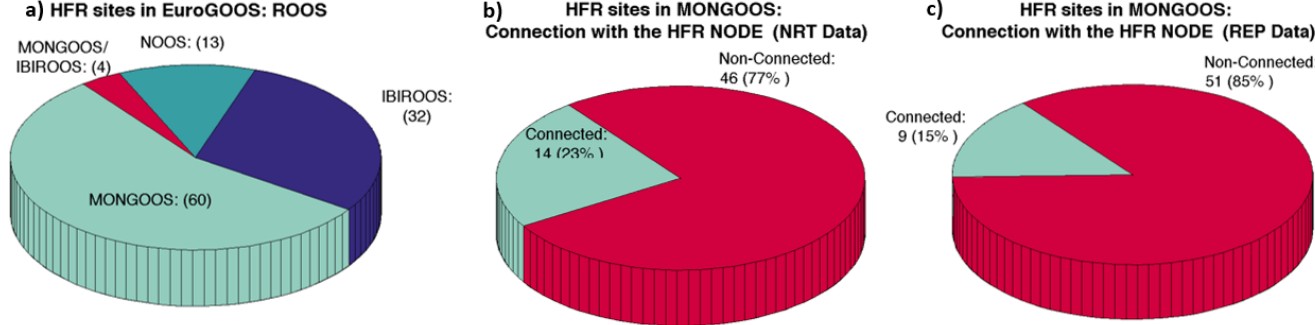


**Figure 9. Pie charts showing the number and percentages of HFR systems in MONGOOS in terms of Regional Ocean Observing Systems (ROOS) alliances and integration of near real time (NRT) and reprocessed (REP) HFR data into the European HFR Node (Fig. 7).**

## 4. Multi-institutional collaborative projects with HFRs in the Mediterranean Sea

The extension and consolidation of a cross-border network of HFRs in the Mediterranean Sea, which is nowadays integrated with other existing oceanic observation infrastructures, constitutes an essential process that has been supported and still is undertaken within the framework of a number of relevant cooperation projects. Some of these multi-institutional projects, which are listed in Table 3 and described below, aim at building synergies among academia, management agencies, state government offices and end users to guarantee a coordinated development of tailored products that meet the societal needs,

serving the marine industry with dedicated smart innovative services, along with the promotion of strategic planning and informed decision-making in the marine environment.

| Project | Period | Funding body | Web link |
|---|---|---|---|
| TRADE | 2010-2013 | POCTEP programme,  ERD funding | www.tradehf.eu/en/home |
| TOSCA | 2010-2013 | MED programme, ERD funding | (There is no active website) |
| RITMARE | 2012-2016 | National research programme funded by the Italian Ministry of University and Research | www.ritmare.it/en |
| HAZADR | 2013-2015 | IPA Adriatic Cross-border Cooperation Programme | https://www.hazadr.eu |





| NEURAL | 2013-2015 | Unity Through Knowledge (UKF) Fund | http://www.izor.hr/neural |
|---|---|---|---|
| JERICO-NEXT | 2015-2019 | H2020 programme INFRAIA | www.jerico-ri.eu/previous-project/jerico-next |
| IMPACT | 2017-2020 | INTERREG Italy-France maritime programme | www.impact-maritime.eu/en/project/ |
| IBISAR | 2018-2020 | CMEMS User uptake programme | www.ibisar.es |
| SICOMAR PLUS | 2018-2021 | INTERREG Italy-France maritime programme | www.interreg-maritime.eu/it/web/sicomarplus |
| SINAPSI | 2020-2022 | INTERREG Italy-France maritime programme | www.lamma.rete.toscana.it/en/progetti/sinapsi |
| CALYPSO CALYPSO-FO CALYPSO-South | 2013-2017 2015 2018-2021 | INTERREG Italy-Malta maritime programme | https://www.calypsosouth.eu/ |
| PANORAMED | 2017-2022 | INTERREG MED programme | http://governance.interreg-med.eu/ |
| SHAREMED | 2019-2022 | INTERREG MED programme | https://sharemed.interreg-med.eu/ |
| EUROSEA | 2019-2023 | H2020 programme BG2019-1 | www.eurosea.eu |
| iWaveNET | 2020-2023 | INTERREG Italy-Malta maritime programme | https://italiamalta.eu/progetti/funded-projects/i-wavenet/?lang=en |

**Table 3. Multi-institutional projects dealing with HFR technology in the Mediterranean Sea. ERD, POCTEP and H2020 stand for "European Regional Development", "Programme of cross-border cooperation between Spain and Portugal" and "Horizon 2020", respectively.**


**TRADE: Trans-regional RADars for Environmental applications (2010-2013)**

TRADE was a cooperative program between Spain and Portugal (POCTEP), supported by European FEDER funding. The project´s main goal was to prevent the risks associated with navigation and port operations in the SW Iberian Peninsula and

the Strait of Gibraltar since this corridor has one of the most intense maritime traffic of oil and chemical tankers. To this end, an HFR system was deployed to monitor currents and waves (Lorente et al., 2014). Complementarily, a border interoperability platform was created for the management and distribution of HFR data.

**TOSCA: Tracking Oil Spills and Coastal Awareness network (2010-2013)**

The 5-country TOSCA pilot project aimed at improving the quality, speed and effectiveness of the decision-making process

in case of marine accidents in the Mediterranean, concerning oil spill pollution and SAR operations (Berta et al., 2014, Bellomo et al., 2015). Among other valuable goals, Lagrangian comparisons of HFR-derived measurements were conducted against the trajectories provided by drifters previously released in high traffic coastal areas to provide critical information to support policy makers.

**RITMARE (2012-2016)**

The Italian flagship project RITMARE focused its efforts on: i) the integration of the existing local observing systems, toward a unified operational Italian framework; and iii) the harmonization of data collection and data management procedures (Carrara et al., 2014). A specific action was conducted for the establishment of a national coastal radar network that included both HFR and X-band radar technologies (Corgnati et al., 2014). Furthermore, a dedicated action was undertaken to foster interoperability among different data providers.

**HAZADR (2013-2015)**

This project aimed to upgrade the knowledge framework about the estimated environmental and socio-economic risks in the most vulnerable areas of the Adriatic Sea, due to both natural and human-induced factors. Furthermore, a decision support system was implemented to track the spreading of oil spilled during hazards. The usage of 6 HFR systems in different applications were part of the project, with some of them (like HFR-Split, shown in Fig. 1) being installed during the project,

while sharing the data in a common database.

**NEURAL (2013-2015)**

The main objective of NEURAL project was to build an efficient, reliable and innovative prototype of ocean surface current forecasting system in coastal areas of the northern Adriatic Sea, by using neural network algorithms. The Self-Organizing Maps (SOM) neural network was trained jointly by the multi-year surface current fields measured by HFR and mesoscale

surface winds simulated by high-resolution numerical weather prediction models. Then, based on the weather forecast and

the trained SOM solutions, the prediction of surface currents was issued for the following three days, to which mesoscale atmospheric models have a significant reliability. The SOM-based forecast was verified against an independent dataset, showing to have slightly higher reliability than the classical ocean forecasting system based on numerical modelling.

**JERICO NEXT (2015-2019)**

The JERICO-NEXT (Joint European Research Infrastructure network for Coastal Observatory - Novel European eXpertise for coastal observaTories) initiative, carried out by 33 institutions from 15 countries, emphasized that the complexity of the coastal ocean cannot be well understood if interconnection between physics, biogeochemistry and biology is not guaranteed. Such integration required new technological developments allowing continuous monitoring of a larger set of parameters. JERICO-NEXT consisted of strengthening and enlarging a solid and transparent European network to provide operational

services for a timely, continuous and sustainable delivery of high-quality environmental data and products related to the marine environment in European coastal seas. In terms of HFR technology, the main aim was not only to harmonize data formats and best practices but also to improve current estimates (by means of advanced quality controls) to study ocean transport and connectivity between coastal and deep open sea waters.

**IMPACT (2017-2020)**

The IMPACT project aimed to establish the first transboundary HFR network between Italy and France, covering 200 km of coastline. The main goal was to define cross-border sustainable management plans to preserve marine protected areas that take into account the development needs of ports, both fundamental elements of the so-called Blue Growth. IMPACT also promoted shared best practices to improve the interoperability and usability of the entire system. IMPACT capitalized investments on HFR technology and constituted the starting point for a further expansion of the network, thanks to

SICOMAR plus and SINAPSI projects, which are also described below in this section.

**IBISAR (2018-2020)**

The IBISAR (Iberia-Biscay-Ireland Search And Rescue) service, implemented within the context of CMEMS User uptake programs, aimed at facilitating decision-making to SAR operators and emergency responders (Révelard et al., 2021; Reyes et al., 2020). IBISAR is a coastal downstream service that provides a user-friendly ocean data quality assessment with easy-

interpretable metrics to guide users to select the most accurate ocean forecast in the IBI region, including the Western Mediterranean Sea, and facilitate decision-making. To this aim, 9 ocean forecast models (4 CMEMS models, 2 regional models and 3 coastal models), 6 HFR systems and all drifters available in the CMEMS catalogue were integrated.

**SICOMAR PLUS (2018-2021)**

The SICOMAR plus cross border Italy-France project addresses the common challenge of navigation safety and quality of

the transboundary marine environment. The project's overall objective is to reduce the risks associated with navigation

accidents and their consequences on human life, goods and the environment. It will create a coordinated system of governance tools, highly technologically innovative surveillance methods and new safety services at sea. The project intends to launch shared strategic planning activities which will identify navigation safety solutions in high-risk marine zones of the cooperation area by setting up two joint monitoring plans for navigation and pilotage safety. The project aims to improve the

coverage of monitoring networks, increase the effectiveness of risk reduction forecasting systems, enhance environmental protection services and establish interoperable data sharing. To this end, several new HFR systems have been installed and some other upgraded along the Italian and French coastlines, respectively (Guérin et al., 2019).

**SINAPSI (2020-2022)**

The SINAPSI project (Assistance to Navigation for Access to Safe Ports), aims to develop real-time tools to monitor the sea

state for safe navigation and wise decision-making in port-approach areas, thereby reducing the risk of accidents. The objective will be pursued by expanding and integrating the cross-border monitoring network of traditional instruments (ADCPs, drifters, etc.) with innovative tools such as coastal HFRs. Additionally, the network will then be used to validate a series of numerical models required for the prediction of the hydrodynamic conditions in port-approach areas.

**PANORAMED (2017-2022)**

PANORAMED is a governance platform that supports the process of strengthening and developing multilateral cooperation frameworks in the Mediterranean region for joint responses to common challenges. The whole Mediterranean space is represented by the 12 member states included in the partnership. Within this timeframe, PANORAMED will provide opportunities to: i) organize high level events aiming at improving the Mediterranean area's governance covering the whole territory; ii) promote the preparation of strategic projects, through dissemination events in each country and the preparation

and launch of the so-called "Terms of Reference". During the first two years, PANORAMED will work on two strategic themes, "Coastal and maritime sustainable tourism" and "Maritime surveillance", with a future extension of "Innovation" as a third strategic theme.

**SHAREMED (2019-2022)**

SHAREMED (SHARing and Enhancing capabilities to address environmental threats in the MEDiterranean sea) focuses on

increasing the capabilities to assess hazards related to pollution and environmental threats in Mediterranean transnational waters. This goal will be achieved by sharing knowledge, observations and technologies as well as building common frameworks, tools and services to evaluate the impact of environmental threats on marine ecosystems. The SHAREMED HFR group aims to enhance the quality and use of HFR observations by merging them with other observational and modelling data sources.





**EUROSEA (2019-2023)**

The project "EuroSea: Improving and Integrating European Ocean Observing and Forecasting Systems for Sustainable use of the Oceans" works to enhance the European ocean observing and forecasting system in a global context, delivering ocean observations and forecasts to advance scientific knowledge about ocean climate, marine ecosystems and their vulnerability to human impacts, demonstrating thereby the importance of the ocean for an economically viable and healthy society. It aims at advancing research and innovation towards a user-focused, truly interdisciplinary, and responsive European ocean observing and forecasting system, that delivers the essential information needed for human wellbeing and safety, sustainable development and blue economy in a changing world. With regards to HFR technology, EuroSea aims to establish the governance structure (Rubio et al., 2021) and the implementation of best practices of operations, including an outage online reporting database, a standardized quality assessment and an effective data management.

**CALYPSO, CALYPSO-FollowOn and CALYPSO-South**

Through the CALYPSO, CALYPSO FollowOn and CALYPSO-South projects, a permanent and fully operational HFR system for the real-time measurement of sea surface currents and waves in the strip of sea between Malta and Sicily was set up (Orassi et al., 2018). Data applications are opened to many different sectors, reaching out beyond research and monitoring, targeting downstream services in support of key national and regional stakeholders. The objective of the 2-year CALYPSO project was the deployment of the HFR system for the permanent monitoring of the sea state. CALYPSO-FollowOn (2015) was a 6-month intensive project which built on the achievements of CALYPSO project. It delivered a more robust HFR monitoring of sea surface currents in the Malta-Sicily Channel with the installation of an additional HFR site on the Sicilian side. CALYPSO-South (2018-2021) currently addresses the challenges of safer marine transportation, protection of human lives at sea, and safeguarding of marine and coastal resources from irreversible damages. It is a commitment to put technological advancement and scientific endeavor at the service of humanitarian responses, reducing risks in sea faring and protecting the marine environment. To this end, the CALYPSO HFR network coverage was expanded to the western part of the Malta-Sicily Channel and the southern approaches to the Maltese archipelago, developing new monitoring and forecasting tools, and delivering tailored operational downstream services to assist national responsible entities in their maritime security, rescue and emergency response commitments.

**iWaveNET (2020-2023)**

iWaveNET aims to implement an innovative network to monitor the sea state along the southwestern coast of Sicily in a cross-border area through the integration of different technologies, encompassing HFR, directional wave buoys, high sensitivity seismographs, tidal gauges and numerical models. The final scope is to develop a Decision Support System to be transferred to interested parties (local and national authorities) for the mitigation of the coastal risk linked to extreme events (i.e., storm surges, etc.) that are potentially catastrophic in the Sicilian channel.



## 5. Future challenges and prospects

### 5.1. General challenges

Equally to other operational ocean observing systems existing in the Mediterranean Sea (for an extensive review, see Tintoré et al., 2019), there are diverse socio-economic and technical challenges to be tackled during the implementation of an
integrated HFR regional network. A SWOT analysis was performed as a situational framework not only to assess the current status and future prospects of this coastal network but also to evaluate strengths, weaknesses, opportunities and potential threats associated with this implementation process that could eventually aid to foster the long-term strategic planning and wise decision-making (Fig. 10). Among others, the top priority issue is not only the maintenance of continued financial support to preserve the infrastructure core service already implemented and subject to costly repairs, but also the pursuit of
permanent funding to extend the network at both national and regional scales for better cross-border coverage. The networks are frequently supported by national research funds and their long-term sustainability is thereby seriously jeopardized. Furthermore, the monitoring capabilities are variable, with a clear north-south unbalance in the Mediterranean region due to the existence of fragile political systems in southern shore countries (Fig. 10).






**Figure 10. SWOT (Strengths, Weaknesses, Opportunities, and Threats) analysis of the Mediterranean HFR network.**

A network extension should fulfill a number of interlinked requirements: (i) simplification of bureaucratic processes for obtaining licenses; (ii) finding and gaining access to suitable and unobtrusive emplacements; (iii) training of new technicians to operate the network, which would include the dissemination of the latest available methodologies to ensure that the most up-to-date best practices are followed; (iv) streamlining the visibility of HFR as a non-invasive remote-sensing technology for maritime surveillance with a broad range of practical applications and subsequent societal benefits. In this context, holding open-house conferences and workshops, not only focused on HFR operator community and permitting agencies but also on a more general non-instructed audience, might be an effective way of promoting public awareness and ensuring the network's survival.

In spite of the fruitful collaborations between the HFR national networks, the coordination and long-term integration at regional scale are sometimes handicapped by poor data policy and restricted data access (Fig. 10). There is still a recognized necessity for the unification of standards, the centralization of methodologies and best practices documentation to increase not only the interoperability of the coastal HFRs network design, operation and maintenance tasks but also the efficient data discovery (Mantovani et al., 2020).



A complementary aspect would be the implementation of an harmonized outage reporting among the HFR community, at both European and Mediterranean levels. This would imply the creation of a centralized HFR outages database (Updyke, 2017) as an ancillary support for operations and maintenance in order to ensure HFR sites sustainability (i.e., downtime,
outages and failures). It would work as a forum to share expertise, integrate approaches and minimize the impact of temporal outages (Roarty et al., 2019).

Additionally, the communication with policy-makers and stakeholders is, even now, occasional and intermittent. Potential stakeholders should be clearly identified and promptly informed to boost their engagement. The success of any regional alliance inexorably relies on the adoption of a win-win strategy, based on transparency, where commitments are both
measurable and achievable by means of well-defined milestones. A bidirectional commitment should be built between HFR operators ("we create the tailored product you urgently need") and stakeholders ("we will definitely use the products and services you specifically implement for us"). Afterwards, tracking and keeping commitments is recognized as one of the most relevant aspects of stakeholder relationship management. Fluent and seamless communications, tracked in a detailed and time-based manner, are essential to update all groups affected over the course of the collaboration. More importantly,
both stakeholder´s needs and/or HFR operators' resources can change along the commitment lifespan so periodic upgrades of the action plan might be required to satisfactorily match each other. In this context, the promotion of successful synergies and "happy-ending stories" might constitute an effective way to attract and mobilize new stakeholders (pre-existing or new-born) by means of the foundational sequence "tell, sell, negotiate, enlist". The EU HFR node or IBISAR project (Révelard et al., 2021) constitute successful examples of this bidirectional long-term engagement between HFR operators and end-users
such as SASEMAR (the Spanish Marine Safety Agency).

Given the broadly accepted credibility of HFRs, this technology must be integrated into robust analysis frameworks for improved marine governance over coastal resources, covering a range of dimensions, such as legislative, planning, infrastructure, technical, scientific and institutional partnerships at Mediterranean level. HFRs can positively contribute to the proper establishment of environmental policies and strategies, bridging the gap between research and societal challenges.

**5.2. Technical challenges**

The last two decades have witnessed the evolution of oceanographic HFR systems from a collection of local and regional instruments operated by research-oriented groups to a back-bone element in emerging national coastal ocean observatories. The practical applications already developed have unequivocally demonstrated that HFR-derived surface currents are a reliable resource for SAR operations, oil spill tracking or harmful algal bloom monitoring, among others. In addition, pilot
programs have been undertaken by national agencies to evaluate the potential ingestion of other HFR basic products such as directional wave and wind information, together with the implementation of ad hoc alert systems for tsunami detection and vessel tracking. All these scientific and operational developments have been key drivers for the steady evolution of HFR technology, which aims to respond adequately to both societal priorities and the growing end-users demands.



A relevant technical challenge that must be faced and successfully overcame over the upcoming years is the resilience of
HFR coastal networks, which is seriously handicapped by harsh met-ocean conditions (i.e., heat, strong wind gusts, salt,
heavy rain and moisture) and the periodic passage of storms that give rise to severe sea states (Medicanes, storm surges and
tsunamis). Served as recent example, the HFR system deployed in the Ebro Delta (NE Spain) was able to provide accurate
and sustained observations during the record-breaking storm Gloria, which hit the NW Mediterranean Sea in January 2020,
proving thereby to be resilient to extreme events (Lorente et al., 2021).

Notwithstanding, resiliency is a broad concept that applies not only to hardware, but also to software. The HF band has been
described as a clutter-rich environment. HFR manufacturers have implemented and keep developing robust software in order
to mitigate clutter from both environmental and anthropogenic sources, including lightning, radio transmissions, ionospheric
echoes and wind turbine echoes, to name a few.

In addition to resiliency, automation in the management of HFR systems is a key element to minimize operating costs at both
national and regional scales and to ensure the long-term sustainability of the network. To meet this need, HFR manufacturers
include a variety of dedicated tools and software packages, developed to operationally monitor radar system health in real
time so abrupt anomalies in some variables (i.e., temperatures, voltage supply levels, forward and backward transmitted
power, among others) or gradual degradation and failure problems can be easily detected, triggering alerts for
troubleshooting. Furthermore, newly developed software, used together with information provided by AIS antenna on the
radar site, allows using the position of "ships of opportunity" to constantly monitor and automatically upgrade the
performance of both DF and BF algorithms. Despite all these available tools, HFRs do occasionally require maintenance
and/or a corrective response, similarly to any other observational network. However, radar operators are often purely
scientific driven and have limited capabilities and resources to cope with this, often affecting the availability and/or quality
of the data obtained.

In addition, weather radar operators´ footsteps should be followed since there is an increasing competition for operating
bandwidth. As HFR broadcast licenses were traditionally issued as secondary, obtaining dedicated frequency allocations has
remained as a priority for a long time. In 2012, the International Telecommunications Union (ITU) officially allocated
frequency bands between 3 and 50 MHz to support HFR operations (Roarty et al., 2019). Notwithstanding, this allocation is
not exclusive, especially in the Mediterranean Sea, and such bands are nowadays used by other official and non-official radio
services. As previously pointed out by Bellomo et al. (2015) in the frame of TOSCA project, acquiring a frequency
allocation that allows HFR as a primary user constitutes a key objective for the Mediterranean community in order to
mitigate the presence of radio frequency interferences that significantly impact on HFR performance. With ITU regulations
becoming increasingly adopted around the world, more and more HFR stations have to share limited, fixed frequency bands.
The expansion of HFR systems in the Mediterranean makes frequency sharing and coordination among different networks of
vital importance.

As a general technical challenge, HFR systems are permanently ameliorated. On one side, the hardware is steadily improved
to minimize space, maintenance tasks and inherent costs. Such improvements include: i) for DF systems, the recent



development of long-range crossed-loop monopole systems on a single mast; ii) for phased-array BF systems, the availability of small low-cost measurement devices that allow for measuring and calibrating cable phases at the electronics rack (no field work required) and the implementation of Multiple Input Multiple Output receive antenna arrays that reduce the antenna footprint without sacrificing performance. On the other side, novel software processing strategies are constantly being developed and updated to improve the quality of the measured data. Such developments encompass: i) for DF systems, a new wave processing software that considers antenna pattern measurement; ii) for phased-array BF systems, software upgrades to apply DF techniques on this type of HFR to improve azimuth precision on far ranges.

Served as a summary, main technological challenges for the upcoming future would encompass: i) improving resilience and automaticity to keep down operating costs; ii) eliminating (or at least reducing) the impact of radio noise and interferences through better enforcement of ITU band utilization and further development of digital filters; and iii) increasing technical readiness level of additional data products (beyond surface currents) via a more direct engagement with stakeholders.

## 5.3. Research challenges

Among the research challenges, integration must be achieved by building reinforced synergies between commercial developers, academic institutions, management agencies and state government offices for a coordinated creation of tailored products to support end-user communities. In this context, HFR-derived products should evolve towards finer spatio-temporal scales to improve the coastal ocean monitoring, in line with the announced CMEMS coastal extension (Sánchez-Arcilla et al., 2021), and thereby resolve adequately littoral (sub)mesoscale processes of paramount relevance. The accurate retrieval of HFR surface currents remains as a top priority since it is a prerequisite of the existing applications of this shore-based technology. The main challenges, already being addressed to properly estimate radial velocities at increased spatial coverage, are related to the correct identification of the first-order Bragg peaks and their exact locations and the resolution of the received signals in range and azimuth. This would aid to fulfil the recommended level of data provision: 80% of the spatial region over the 80% of the time (Roarty et al., 2012).

Complementarily, the accurate monitoring of transport processes also remains as a prime concern due to its influence on SAR operations and oil spill emergencies. Lagrangian time-dependent approaches with HFR data, such as Lyapunov exponents (Nolan et al., 2020) and Lagrangian Coherent Structures (Haller, 2015), provide a robust framework to resolve coherent flow patterns. However, they are often time consuming and computationally more expensive as they require trajectory integrations over a complete spatio-temporal velocity dataset. Since hardware or software failures in the HFR system occasionally compromise the availability of data, diverse methodologies have been proposed to fill spatiotemporal gaps in HFR measurements, encompassing self-organizing maps (SOM), open-boundary modal analysis (OMA) or data interpolating empirical orthogonal functions (DINEOFs), among others. Despite the growing relevance of such approaches, there is still an active debate on the limits of applicability of each gap-filling method for the Lagrangian assessment of coastal ocean dynamics (Hérnandez-Carrasco et al., 2018). Nonetheless, there seems to be consensus about the convenience of combining HFR data with both Eulerian and Lagrangian approaches, when possible, to properly explore transport



processes at (sub) mesoscale ranges. A halfway approach, denominated Eulerian Coherent Structures, has been recently developed to connect Eulerian quantities to short-term Lagrangian transport (Serra et at., 2020), with substantial benefits in SAR operations.

HFR-derived wave parameters are receiving growing attention, but mainly within the academia and research environments.

In terms of operational oceanography, HFR-derived wave data are still far from being used on a near real time basis, in contrast to surface currents which have reached a very mature stage. In order to assess the accuracy of HFR-derived wave data, several validation studies have been carried out in the Mediterranean Sea (Table 1). Results suggest that HFR can efficiently monitor the wave field, even during extreme events when wave heights exceed the predefined saturation limit of the HFR, which depends on the frequency (Lorente et al., 2021). There are diverse challenges associated with the retrieval of

wave parameters that must be still addressed to foster the operational use of this basic product, encompassing the appropriate application of a common battery of automatic checks performed in real time (to flag and subsequently filter inconsistent values or spike-like fluctuations) or the standardization of data and metadata structure. Additional efforts should be focused on the improvement of multiscale wave height estimation for highly variable sea states by using dual-frequency HFR systems (Wyatt and Green, 2009; Helzel et al., 2017) or by extracting wave information directly from the first-order Bragg

peaks (Zhou and Wen, 2015) in order to overcome the wave height limitation at single-frequency and to better measure low and moderate waves, respectively.

Future research endeavors should also include the development of robust algorithms for a reliable measurement of wind speed, which remains less developed, that could complement the ongoing HFR multi-parameter monitoring. The limited number of studies existing in the Mediterranean areas (i.e. Ligurian Sea, Gulf of Naples) about the extraction of ocean

surface wind from HFR systems, seem to suggest that the accuracy of wind field inversion algorithms in coastal areas improves for higher frequency systems under strong wind conditions. They also recommend prior knowledge about the wind field variability and climatology in the study area to better design the investigation and assess the wind field measurements.

On the other hand, the implementation of data assimilation schemes could provide the integrative framework for maximizing the joint utility of HFR-derived observations and numerical models with the aim of improving model predictive skills in

coastal areas. Although few valuable initiatives have been already carried out in the Mediterranean with positive results in the modelling of the upper layer circulation (Hernández-Lasheras et al., 2021; Vandenbulcke et al., 2017; Marmain et al., 2014), we should further strive to develop robust, fully operational assimilation schemes for HFR data, encompassing both radial and total current vectors. Equally, some previous works outside the Mediterranean study area have reported the benefits of assimilating HFR-derived wave parameters into SWAN wave models (Siddons et al., 2009) or the high-resolution

coastal Wavewatch III model (Waters et al., 2013).

Although data assimilation is a powerful technique, advances in coastal ocean monitoring should also include an improved understanding of underlying physical processes. For instance, wave-current interactions can contribute to the generation of large-amplitude waves, triggered naturally when a stable wave train encounters an accelerating opposing current (Onorato et al., 2011). Ràfols et al. (2019) drew a similar conclusion via numerical simulations with coupled (hydrodynamic-wave)





models in the NW Mediterranean Sea. Viitak et al. (2016) reported an increase of the wave height of up to 100 cm in nearshore waters of the eastern Baltic Sea, during the St. Jude storm, due to the combined effect of surface currents and sea level on the wave field evolution. In this context, HFR technology should be thereby used to effectively monitor extreme events in near real-time and unveil hydrodynamic aspects such as the aforementioned wave-current interactions (Zeng et al., 2019), which are still poorly resolved or even misrepresented by current state-of-the-art regional ocean models (Lorente et

al., 2021).

## 6. Summary and conclusions

Over recent decades, HFR has become commonplace in monitoring the sea state in coastal areas, once its technical capabilities and potential applications have been clearly showcased. With the maturing of this technology, attention has turned to what the scientific community and other end-users can learn and build-up from HFR data.

Since the Mediterranean Sea constitutes a first order geostrategic region from both commercial and oceanographic perspectives, the use of HFR has been steadily gaining recognition as an effective land-based remote sensing technology for the multi-parameter monitoring of the socioeconomically vital and often environmentally stressed coastal waters. The present work is intended as a panoramic overview of the main achievements, ongoing activities and future challenges to be faced by the Mediterranean HFR community in order to transition several standalone HFR systems into an integrated

monitoring network, operated permanently at basin scale. While the implementation of a fully operational HFR regional network in the Mediterranean Sea is still in progress and far from complete, the pragmatic lessons already learned and application examples here illustrated might be useful to similar programs under development elsewhere.

A detailed description of the roadmap adopted to transform individual radars into an integrated HFR network has been provided. To assess the maturation process into a fully operational status, the system must evolve via an implementation of

phased approaches, including: harmonization of HFR systems architecture, homogenization of deployment and good practices for preventive maintenance, data format convergence (i.e., standardization of files structure, metadata and automatic quality control tests), regular validation exercises against independent in situ observations, centralization of data management and access platforms, and eventually the development of customized visualization tools and added-value products to facilitate data discovery.

While this paper constitutes the introductory part of a double contribution where the current state-of-the-art is thoroughly presented, the second part addresses the latest scientific breakthroughs with HFR technology achieved in the Mediterranean region to fulfil stakeholders demands (Reyes et al., submitted to this Special Issue). In particular, the second manuscript is built over three main cornerstones (maritime safety, extreme events monitoring and ecological decision support) to showcase emerging research-based downstream applications of societal benefit founded on the operational use of quality-controlled

HFR-derived data.



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

**Author contributions**

PL and ER conceived the idea of this manuscript and fostered the collaboration as MONGOOS-HFR co-chairs, being in charge of overall direction and planning. All authors provided inputs for Sect. 3 and 4 and contributed to the writing of other sections, as follows: PL took the lead in writing the abstract and preparing the first draft of the document. RG and JS contributed significantly to Sect. 2 and 5. CAG, AM, MB, MBen, MM, AG, RS, DD, EZ, PF, ACE, IHC, JHL, ST, EA, CB, BD, MF, HM, VD, ML, LU, BM and AR contributed to section 2.2. YT, FC, GC, AD and AGau contributed to Sect. 2.2.2. EZ, DC, SS, PF and MU contributed to Sect. 2.2.2 and 2.2.3. LC and CM contributed to Sect. 2.3, 2.4, 2.5 and 2.6. Finally, JT, IV, CAG, AM, AO, LU and VC conducted a detailed revision of the entire manuscript to gain cohesion.

**Acknowledgements**

This study has been developed partially in the framework of the Interreg MED Strategic Project SHAREMED, co-financed by the European Regional Development Fund under the Funding Programme Interreg MED 2014–2020. Website: https://sharemed.interreg-med.eu/. We also thank the partial support of the CMEMS-INSTAC phase II, which provides the context of the activities for HFR data harmonization, standardization and distribution. Collaborative discussion on data



management harmonization at European level has been carried on also thanks to the contribution of the projects INCREASE (CMEMS Service Evolution Call for Tenders 21-SE-CALL1) and SeaDataCloud (EU-H2020 GA n. 730960).

It is also worthwhile mentioning that the present work has been possible thanks to MONGOOS collaborative network, aimed toward long-term synergies between multi-disciplinary working groups in the Mediterranean Sea in order to launch strategic


initiatives and pursue funding for innovative research projects.

Finally, authors would like to express their gratitude to the internal reviewers, Anne Molcard, Ivica Vilibić, Charles-Antoine Guèrin and Joaquín Tintoré and to the MONGOOS co-chairs, Vanessa Cardin and Alejandro Orfila, for their careful and meticulous reading of the manuscript. Their detailed and comprehensive reviews have been very helpful to improve the structure, the reading and to finalize the manuscript.

**Code availability**

Corgnati, L. (2020, May 26). LorenzoCorgnati/HFR_Node_tools: EU_HFR_NODE_Tools (Version v2.1.2). Zenodo. http://doi.org/10.5281/zenodo.3855461

Corgnati, L. (2019, December 10). LorenzoCorgnati/HFR_Node_Historical_Data_Processing:

EU_HFR_NODE_Historical_Data_Processing (Version v2.1.1.6). Zenodo. http://doi.org/10.5281/zenodo.3569519

Rotllan, P. (2021, June 2). High Frequency Radar data visualization (Copernicus Marine Service), https://doi.org/10.17882/80874.

**Data availability**

Standardized HFR data is available in the Thredds Server from the European HFR Node for some of the HFR systems mentioned in this research (http://150.145.136.27:8080/thredds/HF_RADAR/HFradar_CMEMS_INSTAC_catalog.html)

MIO's HFR-Toulon data, used in Sect. 2 (Fig. 3) are available at http://hfradar.univ-tln.fr/HFRADAR/squel.php?content=accueil and real-time total currents (hourly data) are available for 2020 and 2021 in https://erddap.osupytheas.fr/erddap/files/cmems_nc_cf0e_c84a_8ead/

In situ observations provided by a moored buoy (from Stazione Zoologica Anton Dohrn Napoli), used in Sect. 2 (Fig. 5), are available at https://www.szn.it/index.php/en/research/research-infrastructure-for-marine-biological-resources/access-to-

marine-ecosystems-and-environmental-analysis/infrastructure-for-marine-research-irm/meda-b-napoli

In situ observations provided by a moored buoy (from Puertos del Estado), used in Sect. 2 (Fig. 4 and Fig. 5), are available at https://www.puertos.es/es-es/oceanografia/Paginas/portus.aspx



**Competing interests**

Author JS currently is employed at Qualitas Instruments Lda. Author RG currently is employed at HELZEL Messtechnik
GmBH. Author PL currently is employed at NOLOGIN Consulting SL. However, these three authors have not advertised
commercial products and the research has not been sponsored by any one of the companies.

AO and VC are guest members of the editorial board of the Special Issue from the Journal. The peer-review process was
guided and overseen by another member of the editorial board.

The remaining authors declare that there are no relevant financial or non-financial competing interests to report.





