# Peer review of "Coastal HF radars in the Mediterranean: status of operations and a framework for future development"

_Ocean Science, 2021_

## Referee Comment (RC1)

General Comments

The paper provides a good overview of High Frequency radar and how it can be used to answer scientific and societal questions related to the coastal ocean.  They then discuss the measurements of HFR (currents, waves and winds).  They then document the long list of projects that have contributed to the development of HFR in the Mediterranean.  Lastly, they lay out the future challenges and prospects for HFR technology in the region.

Specific Comments

I would like to hear from the authors on how HFR compares to other measurements in the Mediterranean in terms of priority and significance.  Also, can the authors document more "happy ending stories" line 957.  Can they explain more on SASEMAR?  Are there other stories like this one?

Technical Comments

| | |
|---|---|
| Line 53 | Can the authors expand on what they mean by Mediterranean framework, do they mean region?  If not please expand and clarify |
| Line 58 | what is meant by a restless navigable route |
| Line 60 | remove the such as, there are only 3 chokepoints just state the 3 |
| Line 63 | can the biological productivity be mapped in Figure 1 rather than bathymetry? |
| Line 75 | It is hard to the see the dots representing the HFR stations, can they be made larger? |
| Line 79 | 7% of the worlds biodiversity while only covering 0.7% of the ocean's surface area |
| Line 90 | "impact assessment" can the authors expand upon this phrase? |
| Line 99 | 470 million |
| Line 105 | enhanced maritime safety and improved ecological decision support sounds like US IOOS challenges.  Are these challenges documented in Europe as well? |
| Line 110 | remains a priority |
| Line 124 | for societal benefit |
| Line 125 | can you add a sentence joining this paragraph with the previous eg "For instance, two programs that contribute to the wealth of data collected in the Med include SMOS and ARGO. " |

Line 134      areal and endurance coverage

Line 136      thereby providing

Line 157      the title of the section is Mediterranean Oceanographic Network, line 185 uses
              the term Mediterranean observing system.  Any signifigance in the difference in
              terms?
Line 166      what is meant by boots?

Line 183      Should that read MONGOOS HFR Network?

Figure 2      what is the meaning of the double sided arrow between mongoos and Med HFR
              working group?

Line 186      coastal observing systems should not be capitalized

Line 211      consider removing "not only" and "but also"

Line 240      remove "on"

Line 241      add "the Geometric Dilution…"

Line 250      remove very

Line 253      meaning of ADC?

Line 290      change to ionospheric

Line 300      For this section would you consider a figure like this

|              | Beam Forming | Direction Finding |
|--------------|--------------|-------------------|
| Phased Array | WERA         | WERA              |
| Compact      | ?            | CODAR SeaSonde    |

Line 319      directions of arrival

Line 321      consider this as a good reference for this sentence
              B. M. Emery, "Evaluation of Alternative Direction-of-Arrival Methods for Oceanographic HF
Radars," in *IEEE Journal of Oceanic Engineering*, vol. 45, no. 3, pp. 990-1003, July 2020, doi:
10.1109/JOE.2019.2914537.

Line 322      consider using phased-array rather than extended arrays

Line 323      what is meant by "peaky"?  Can a better term be used or more explanantion
              given?

Line 324      remove very

Line 325      remove careful

Figure 3      do the plots cover the same period?

Figure 4b     remove x axis 45-180 degrees and wrap 0-45 next to 360.  The resolution of the
              figure is poor, increase it.

Line 390      Lagrangian

Line 399      please provide reference for these statistics

Line 407      have also been documented

Line 408      please provide a summary sentence or paragraph on this section, introducing the
              next section

Line 445      new paragraph after reference

Line 453      asset

Line 489      what is Mc-WAF?

Line 490      please explain intermediate rings more

Line 515      location-specific

Figure 5      missing x axis tick marks on the figures, standardize the colors HFR is red, buoy
              blue, other black.  Fig 5d is faint.

Line 537      please provide a summary sentence or paragraph on this section, introducing the
              next section

Line 553      remove also

Line 557      why is this sentence needed?

Line 570      provided

Line 572      showed that

Line 576      should tense be past, showed comparisons

Line 585      and iii) other

Figure 6      cc should be subscript to match figure.  What is the red dashed line in subplot a?
              The RMSE of 60 degrees seems high, please explain.

Line 599      replace know-how with operational experience

Line 600      replace minor with reduced

Line 605      Is Med HFR group working with Ocean Best Practices
              https://www.oceanbestpractices.org ?

Line 608      does the numbering in the list denote a hierarchy?  if not replace with symbols.

Line 617      is there a reference for the EuroGOOS HFR Task Team?

Line 622      consider removing "for the development of operational ocean monitoring via
              HFR systems"

Line 625      spell out EU, is there a reference for JERICO-NEXT?

Line 627      replace within with "with"

Line 631      which set the basis

Line 635      remove "the achievement of"

Line 637      can the authors comment how the European standard format compares to the
              American standard?  Are there any differences?

Line 638      spell out or explain SDC and INSPIRE

Line 639      replace "to be mandatorily" with "are consistently applied to HFR data as
              defined by the EuroGOOS"

Line 646      is there a link for the EU HFR Node that can be provided

Line 660      can anything be said about data users, statistics on data use?

Line 668      does this rule apply to only HFR or all ocean measurements?  please explain

| Line 697 | should the items in the list be capitalized? |
|---|---|
| Line 740 | replace inexistent with nonexistent |
| Line 745 | replace Served as with For |
| Table 2 | spell out PdE and MIO, consider adding a column for the country and removing country from the Institution column.  Should there be an order for the table?  Alphabetical by country? frequency?  Authors can decide. |
| Figure 8c | should Phased array be replaced with beam forming if comparing to direction finding? |
| Figure 9b | the 14(23%) is cutoff at the bottom, please correct |
| Line 785 | serve the marine industry |
| Table 3 | INTERREG is mentioned often, can this be explained.  Would the authors consider replacing the table with a Gantt chart to show the information?  https://pythonawesome.com/a-convenient-but-aesthetic-way-of-creating-a-gantt-chart-with-python/ |
| Line 792 | the first two project have the acronym explained in the title, the remaining ones do not.  Please be consistent. |
| Line 840 | remove "which are also" and "in this section" |
| Line 850 | The project's overall objective was |
| Line 890 | in title it is CALYPSO-FollowOn, on the next line the – is missing?  Also missing years of the project |
| Line 910 | can you add a summary sentence or paragraph on this section, something to tie together all the projects? |
| Line 913 | Equal to other |
| Line 915 | remove "not only" |
| Line 916 | move "strengths, weaknesses, opportunities and threats" to beginning of sentence.  remove potential. |
| Line 920 | pursuit of permanent funding like those of other programs? |

Line 930      what kind of licenses?  radio frequency?  equipment installation?  please state

Line 950      For this paragraph, is this process happening for other measurements?  if so please list.

Line 957      replace happy-ending with success

Line 969      Additionally pilot

Line 982      to mitigate noise and clutter

Line 988      can easily be detected, triggering alerts for operators

Line 989      spell out AIS,  reference
              https://hfradar.msi.ucsb.edu/brian_emery/files/reports/2013_sbir_phase2_final_report.pdf

Line 1016     replace "keep down" with "reduce"

Line 1046     remove very

Line 1084     replace "build-up" with utilize

Line 1085     what is meant by first order, can this be further explained?

Line 1092     examples illustrated here might

---

## Referee Comment (RC2)

Review of Coastal HF radars in the Mediterranean: status of operations and a framework for future development, P. Lorente et al.

This paper is an ambitious community work aiming to showcase the current status of the Mediterranean HFR network and the future roadmap for coordinated actions that will allow this to play a major role in the high-level challenges of the ocean observing landscape in the Mediterranean Sea.

Significant innovations are described, with interesting multi-site approaches and covering a very wide spectra of fields in the overall value-chain from the HFR systems operations to the transfer of advanced data products.

The presented work is also gathering a complete review of the main levers that the community is tackling (BPs, Harmonization, Data Quality, New parameters…) for promoting exchanges between operators, and creating synergies and added value by transforming a set of individual radars into an integrated network.

The description of the community status, difficulties, key references and challenges derives to a very useful roadmap for the current actors of the network, also for the potential future contributors, and in general for the ocean observing community.

The established regional roadmap is well linked to the European and Global initiatives. Some regional specificities are well described, in particular in the SWOT analysis. However, it may be clarified which of those challenges for future development is really answering a specific or prioritized issue for the Region, and which are shared with the European or Global community.

The manuscript will definitely represent an important step forward for the ocean observing community.

Some detailed minor changes and recommendations for improving the manuscript are listed below:

l.138 General capabilities of HFR are explained, but here or elsewhere in the paper, there is no mention to the limitations of using Long Range in semi-enclosed seas like the Mediterranean Sea (difference with the IBIROOS area where there is a significant number of Long-Range systems). This can be also mentioned later in the SWOT Weaknesses or at least as a factor for achieving the full coverage.

Fig.2 A Mediterranean HFR Working Group is mentioned. In the text l.184, it is called Mediterranean HFR network. For consistency, this would be named in Fig2 (not to be confused with Observation Working Group, one of the 3 MONGOOS WGs).

l.200 Errata: the EuroGOOS HFR **T**ask Team (word order and upper case for Task)

l.237: It may be more precise to say 0.5-**5**m in "operating at specific frequencies within the 3–30 MHz band and providing radial measurements which are representative of current velocities in the upper 0.5–**2** m of the water column. See Rubio et al. 2017

In *2.1 Fundamentals of HFR technology*, it could have been mentioned how the common sea states of a semi-enclosed sea like Med may impact the performance of longer range HFR, with

possible consequences to be taken into account in the plan for achieving a full coverage of the Mediterranean coastline.

l.426. The abbreviation GoN is used before the full version that appears l.456.

Table 1. For consistency, I would recommend to use "Gulf of Naples" in Table 1.

l.448: Long et al 2011 (Central California) should not be included in references on European waters.

l.459: using two "alternative data sources" rather than "different platforms"

l.465 errata: against

Figure 5a. Quality of graphs should be improved (size of labels, same limits for Axes Y, same type of lines, general image sharpness).

l.569: Add a comma: In the Ligurian Sea experiment**,**

l.571: adda comma: During the experiment**,**

Section 2.4: the beginning of Section 2.4 appears too general for this section and may better fit in section 1.5. Only aspects dealing with data flow may be kept in the historical review of the roles of the different initiatives.

L636-642: The text may be simplified here as it is a bit redundant with what is explained later in 2.5 and 2.6.

l.651 Figures by mid-2021 should be presented as a result, not as an objective. Or the date and corresponding objective should be updated.

4. Multi-institutional collaborative projects with HFRs in the Mediterranean Sea: I would suggest to mention the ongoing JERICO-S3 and JERICO-DS projects, part of the JERICO-RI initiative. Their impact could be significant in terms of integration of HFR among key coastal observing technologies.

l.923: I would suggest expressing differently the reason for a clear north-south unbalance in the Mediterranean region. The MOONGOOS community knows better than me the regional variability. More than the Political systems themselves, the factors may lie in the differences between environmental policies, resources dedicated to marine monitoring and research programs, socioeconomical and political priorities, etc.

Figure 10. the different Threats may be organized with "Insufficient adoption of HFR currents standardization" after other linked issues like "Lack of agreement on the data policy".

In *5. Future challenges and prospects:*

As part of the roadmap to transform individual radars into an integrated network, a phased approach is proposed. The different steps aim to optimize and consolidate the network of existing systems. However, the network is currently covering "a small portion of the entire coastal domain". As part of this roadmap, authors could add plans for:

- Defining a quantitative objective (number of systems, surface covered...) as a long-term target

- Agreeing a joint methodology to define priority areas at regional level in the development of the network (for example as introduced in JERICO-NEXT Deliverable D3.4)
- Performing coordinated actions towards stakeholders (for example towards National GOOS Focal points who are serving in EOOS Operation Committee)

Additionally, all the general, technical and Research aspects are relevant for the Med and beyond at European and Global level. But it would be interesting to mention a bit more explicitly how will be tackled some regional specificities exposed in the SWOT analysis.

---

## Author Response (AR1)

**REVIEWER #1:**

**GENERAL COMMENTS**

**The paper provides a good overview of High Frequency radar and how it can be used to answer scientific and societal questions related to the coastal ocean. They then discuss the measurements of HFR (currents, waves and winds). They then document the long list of projects that have contributed to the development of HFR in the Mediterranean. Lastly, they lay out the future challenges and prospects for HFR technology in the region.**

Many thanks to the anonymous reviewer for the number of useful comments that will help to significantly improve the quality of the final version of this manuscript. Please find below detailed answers to each comment.

**SPECIFIC COMMENTS**

**I would like to hear from the authors on how HFR compares to other measurements in the Mediterranean in terms of priority and significance. Also, can the authors document more "happy ending stories" line 957. Can they explain more on SASEMAR? Are there other stories like this one?**

We have added the following paragraph at the end of section 4:

"*While the implementation of a fully operational HFR regional network in the Mediterranean Sea is still in progress, other observational networks have reached a very mature stage in terms of number of permanent devices, length of recorded time series and consistency of the quality-control protocols adopted. According to Tintoré et al. (2019), there are 58 buoys capable of measuring waves (most of which are directional), 100 sea level stations, 37 operational current meters, 113 stations monitoring the sea water temperature, 50 salinity stations and 78 Argo-floats in the Mediterranean Sea. In terms of priority and significance, the HFR network might be considered as a useful ancillary tool that complement in-situ platforms, which nowadays constitute a sound monitoring core in this region. Special emphasis has been recently placed on other emerging technologies, such as glider facilities or biogeochemical Argo floats, thanks to their ability to monitor the three-dimensional water column. However, they are not as broadly used as HFRs and the level of operational implementation still remains in a preliminary research phase.*"

Regarding the "happy-ending stories", we have not documented any other story apart from the one related to SASEMAR. We are currently involved in the development of an HFR-derived coastal upwelling index (see Lorente et al. 2020 for further details) that could be useful for a broad variety of end-users dealing with marine resources, water quality, fisheries and aquaculture production. Since this is a work in progress, we consider that it is not convenient to be mentioned explicitly in the manuscript.

With regards to SASEMAR, this agency is among the most significant end-users of reliable met-ocean information in Spain and it has played a key role in the development of the IBISAR service. SASEMAR acknowledges the efforts made in the context of IBISAR to advance towards the development of ocean model skill assessment services, addressing one of their overarching concerns: the accuracy of the predictions.

We have explained a little bit more on SASEMAR at the end of section 5.1.:

*"SASEMAR oversees maritime traffic control, SAR operations, marine environmental protection and training in Spain. In this context, HFR estimations are readily ingested by the Environmental Data Server (EDS) managed by SASEMAR to enhance the emergency planning process for a prompt response."*

Reference:

Lorente, P.; Piedracoba, S.; Montero, P.; Sotillo, M.G.; Ruiz, M.I.; Álvarez-Fanjul, E. Comparative Analysis of Summer Upwelling and Downwelling Events in NW Spain: A Model-Observations Approach. Remote Sens. 2020, 12, 2762. https://doi.org/10.3390/rs12172762

**TECHNICAL COMMENTS**

**Line 53 Can the authors expand on what they mean by Mediterranean framework, do they mean region? If not please expand and clarify**

Not only the region but also a wealth of inherent factors associated with this marginal sea. We wanted to mean the supporting observational structure, budget, scientific and societal priorities along with specific handicaps and environmental threats in the Mediterranean waters that are used to ultimately define the strategy and roadmap of the MONGOOS HFR team.

We have expanded on what we wanted to mean by Mediterranean framework at the end of section 1.6, not in line 53 since we would like to keep the abstract short and concise.

*"In this context, the Mediterranean framework should be understood as the ocean observational infrastructure already implemented together with a range of thematic areas (gaps, resources, dilemmas, strategic issues, etc.) that inherently shape coordinated efforts and the future roadmap of the MONGOOS HFR team."*

**Line 58 what is meant by a restless navigable route**

We wanted to highlight the significantly high number of ships navigating across the Mediterranean since ancient times. We have replaced "restless" by "busy" to avoid any misunderstanding.

**Line 60 remove the "such as", there are only 3 chokepoints just state the 3**

Done.

**Line 63 can the biological productivity be mapped in Figure 1 rather than bathymetry?**

Yes, mass concentration of chlorophyll-a (CHL-a) in sea water can be mapped as a proxy of primary productivity. However, we would prefer to keep the bathymetry map to highlight that HFR systems are coastal platforms.

A map of concentration of CHL-a (expressed in mg/m3), derived from a 2-year (2020-2021) satellite dataset from the Copernicus Marine Service is provided below. As stated in the manuscript, offshore waters exhibit extremely low biological productivity, with the concentration of nutrients decreasing from NW to SE.

[Figure]

This map could be added to Figure 1 as a second panel (b), we leave it to the discretion of the anonymous reviewer.

**Line 75 It is hard to see the dots representing the HFR stations, can they be made larger?**

Figure 1 has been remade to enhance the visibility of the dots.

[Figure]

**Line 79 7% of the worlds biodiversity while only covering 0.7% of the ocean's surface area**

Included in the text.

**Line 90 "impact assessment" can the authors expand upon this phrase?**

We have added a sentence to better clarify the concept of "impact assessment":

"*In this context, impact assessment should be understood as the analysis of the primary metocean factors that give rise to severe coastal disasters and the comprehensive evaluation of the environmental effects on marine resources along with other inherent societal and economic consequences with the final aim of implementing strategic preparedness policies that could reduce both exposure and coastal vulnerability.*"

**Line 99 470 million**

Done!

**Line 105 enhanced maritime safety and improved ecological decision support sounds like US IOOS challenges. Are these challenges documented in Europe as well?**

Yes, maritime safety is a challenge also in Europe, particularly in the Mediterranean Sea as it is one of the world's busiest shipping lanes of oil and gas tankers, container vessels and ships. Furthermore, the recent exploration and exploitation of the hydrocarbons in the Eastern Mediterranean Levantine Basin (Tintoré et al., 2019) involves a higher risk of marine oil pollution. The development of ecological decision support systems has also been prioritised in the context of RADMED monitoring programme (López-Jurado et al., 2015).

Some modifications have been introduced in the manuscript:

"*other interconnected societal challenges in the Mediterranean Sea have been documented (Tintoré et al., 2019; López-Jurado et al., 2015) and include:*"

References:

Tintoré, J., Pinardi, N., Álvarez-Fanjul, E. et al.: Challenges for Sustained Observing and Forecasting Systems in the Mediterranean Sea, Front. Mar. Sci., 6, 568, 2019.

López-Jurado, J. L., Balbín, R., Alemany, F., Amengual, B., Aparicio-González, A., Fernández de Puelles, M. L., García-Martínez, M. C., Gazá, M., Jansá, J., Morillas-Kieffer, A., Moyá, F., Santiago, R., Serra, M., and Vargas-Yáñez, M.: The RADMED monitoring programme as a tool for MSFD implementation: towards an ecosystem-based approach, Ocean Sci., 11, 897–908, https://doi.org/10.5194/os-11-897-2015, 2015.

**Line 110 remains a priority**

Done.

**Line 124 for societal benefit**

Done.

**Line 125 can you add a sentence joining this paragraph with the previous eg "For instance, two programs that contribute to the wealth of data collected in the Med include SMOS and ARGO."**

Done.

**Line 134 areal and endurance coverage**

Included in the text.

**Line 136 thereby providing**

Done.

**Line 157 the title of the section is Mediterranean Oceanographic Network, line 185 uses the term Mediterranean observing system. Any significance in the difference in terms?**

No, there is no difference between both terms. Since the acronym MONGOOS stands for "Mediterranean Oceanography Network for Global Ocean Observing System", we have changed (for consistency reasons):

1) "Mediterranean observing system" by "Mediterranean oceanography network" in line 185
2) "Mediterranean oceanographic network" by "Mediterranean oceanography network" in the title of the section 1.5

**Line 166 what is meant by boots?**

It is a typo: "boots" has been replaced by "boosts"

**Line 183 Should that read MONGOOS HFR Network?**

No, it should not. The entire MONGOOS network includes 3 different sub-groups: 1) observations, 2) modelling tools and 3) applications. The MONGOOS HFR network is a specific component inside sub-group 1 (observations).

**Figure 2 what is the meaning of the double-sided arrow between MONGOOS and Med HFR working group?**

We wanted to mean that there is a two-way interaction between both elements. On one hand, MONGOOS sets the objectives and the strategy for its three main components (Observation, Modelling and Application working groups) and specific subgroups. On the other hand, the Mediterranean HFR working group, as a subgroup, provides data and feedback to better define the strategy (also contributing to carry out the specific actions at regional level), always aligned with the general roadmap established by the EuroGOOS HFR Task Team.

A piece of text has been added to the caption of Figure 2:

*"The double-sided arrow between MONGOOS and the Mediterranean HFR working group intends to highlight the two-way interaction existing between both entities: the former sets specific tasks and the general strategy, while the latter provides data and support to update the predefined roadmap."*

**Line 186 coastal observing systems should not be capitalized**

OK, modified.

**Line 211 consider removing "not only" and "but also"**

Done.

**Line 240 remove "on"**

Done.

**Line 241 add "the Geometric Dilution…"**

Added "the".

**Line 250 remove very**

Done.

**Line 253 meaning of ADC?**

The acronym ADC stands for "analog-to-digital converter", as it has been included in the text.

**Line 290 change to ionospheric**

Done.

**Line 300 For this section would you consider a figure like this**

Many thanks to the reviewer for the recommendation but we would rather prefer to focus on differentiating the two major types of HFR systems (e.g. Beam Forming -BF- and Direction Finding -DF-) without referring to brands or specific private companies. Indeed, the first version of the draft included a table rather similar to the proposed one. Then we decided to remove any mention of the commercial brands (whose representatives are included as coauthors of this manuscript) to avoid any potential conflict of interests between them.

|                | Beam Forming | Direction Finding |
|----------------|--------------|-------------------|
| Phased Array   | WERA         | WERA              |
| Compact ?      | ?            | CODAR SeaSonde    |

**Line 319 directions of arrival**

Done.

**Line 321 consider this as a good reference for this sentence**

**B. M. Emery, "Evaluation of Alternative Direction-of-Arrival Methods for Oceanographic HF Radars," in IEEE Journal of Oceanic Engineering, vol. 45, no. 3, pp. 990-1003, July 2020, doi:10.1109/JOE.2019.2914537.**

Reference added.

**Line 322 consider using phased-array rather than extended arrays**

Done.

**Line 323 what is meant by "peaky"? Can a better term be used or more explanation given?**

We wanted to mean the opposite of smooth. Maybe "irregular" aspect could be more convenient, as indicated in the text.

**Line 324 remove very**

Done.

**Line 325 remove careful**

Done.

**Figure 3 do the plots cover the same period?**

Yes, both are a snapshot corresponding to September 1st, 2020, 06:00 UTC, as stated in the caption.

**Figure 4b remove x axis 45-180 degrees and wrap 0-45 next to 360. The resolution of the figure is poor, increase it.**

Done. See below the new Figure 4:

[Figure]

**Line 390 Lagrangian**

Done.

**Line 399 please provide reference for these statistics**

Added: "*Cosoli et al., 2010; Berta et al., 2014; Lorente et al., 2014, 2015 and 2021; Corgnati et al., 2019a; Lana et al., 2016; Kalampokis et al., 2016; Capodici et al., 2019; Guérin et al., 2021, Molcard et al., 2009; Bellomo et al., 2015*"

**Line 407 have also been documented**

Done.

**Line 408 please provide a summary sentence or paragraph on this section, introducing the next section**

The following paragraph has been added:

*"Once HFR has proved to be a valid instrument to accurately monitor surface currents with high spatio-temporal resolution over wide coastal areas, the ability of this remote-sensing technology to measure waves and wind direction must also be assessed, as detailed in the next two sections."*

**Line 445 new paragraph after reference**

Done.

**Line 453 asset**

Done.

**Line 489 what is Mc-WAF?**

The McWaf system, operating in ISPRA since 2012, provides sea state forecasts over the Mediterranean Sea and over selected Italian regional and coastal areas (Orasi et al., 2018).

A sentence has been added to the manuscript to clarify this point.

**Line 490 please explain intermediate rings more**

In the case of CODAR systems, HFR wave data are retrieved from a number of individual range rings, which extend radially from an origin at the onshore radar site to a certain distance offshore. Therefore, intermediate rings are those placed at intermediate distances from the coast, nor the first one (closest to the HFR site) nor the outermost range ring. This has been clarified in the text.

**Line 515 location-specific**

Done.

**Figure 5 missing x axis tick marks on the figures, standardize the colors HFR is red, buoy blue, other black. Fig 5d is faint.**

Corrected. See below the new Figure 5:

[Figure]

**Line 537 please provide a summary sentence or paragraph on this section, introducing the next section**

We consider this suggestion already solved in response to the reviewer's request in line 408.

**Line 553 remove also**

Done.

**Line 557 why is this sentence needed?**

Removed from the manuscript.

**Line 570 provided**

Done.

**Line 572 showed that**

Done.

**Line 576 should tense be past, showed comparisons**

Done.

**Line 585 and iii) other**

Done.

**Figure 6 cc should be subscript to match figure. What is the red dashed line in subplot a?**

Corrected. The red dashed line shows the circular correlation coefficient threshold (0.5), since values of ρcc> 0.5 indicate a reasonable correlation between the measurements (Saviano et al., 2021). A clarification has been added to the caption of Figure 6.

See below the new Figure 6:

[Figure]

**The RMSE of 60 degrees seems high, please explain.**

Although the RMSE values for wind direction recorded in the Gulf of Naples (SW Italy) seem to be high, they are in line with similar experiments carried out previously in the Mediterranean Sea (Shen and Gurgel, 2018). The limitations in wind retrieval in this study (performed in Saviano et al., 2021) for direction-finding HFR are comparable to those reported for beam forming systems in the Ligurian Sea (NW Italy) under similar atmospheric conditions (Shen and Gurgel, 2018).

In the Table attached below (extracted from Shen and Gurgel, 2018), it can be observed that the RMSE emerged in the range 20-84 cm/s, with lower RMSE values obtained when the wind speed is higher, in accordance with the results exposed by Saviano et al. (2021)

**Table 5.** Comparison of the RMS error of the wind direction related to wind speed using the pattern-fitting and LSM methods in the Ligurian Sea experiment.

| | RMS Error for Wind Direction Measurements (°) | | | |
|---|---|---|---|---|
| Comparison of inversion method | Wind speed (m/s) U > 3 | Wind speed range (m/s) | | |
| | | 0 < U ≤ 3 | 3 < U ≤ 10 | U > 10 |
| Pattern-fitting method | 57.2 | 80.3 | 57.6 | 20.4 |
| LSM method | 65.1 | 84.6 | 64.7 | 24.1 |

The results show that several factors contribute to the accuracy of HFR wind measurements, the operational frequency being the most relevant one. When the radar operates at higher frequencies, the resonant Bragg waves have a shorter wavelength being more sensitive to changes in the wind field. Therefore, a HFR with a higher operating frequency leads to estimations that are in better agreement with in-situ measurements. Another important factor is wind speed. Different studies comparing HFR and in-situ measurements concluded that the retrieval of the wind direction is not reliable under wind speeds below 5 m/s (Lipa et al., 2014).

In the case of Saviano et al (2021), there are other critical physical effects, such as wind duration and fetch, the assumption of direction homogeneity along the range cells, or the adjustment of wind measurement from 10 m above mean sea level to sea surface, that should be included in the evaluation of the radar inversion performance

In conclusion, we have added a paragraph to the text in order to better clarify why the RMSE values here obtained are higher than those obtained from the comparison of other HFR-derived parameters like sea surface currents or the significant wave height:

"*Although the RMSE values obtained for wind direction in the GoN appear to be high, they are in line with similar experiments carried out previously in the Mediterranean Sea (Shen and Gurgel, 2018). Detected differences could be, in part, attributed to a number of relevant elements such as: (i) sensors' limitations (and the related instrumental noise); (ii) mismatch in the horizontal sampling (whereas direction homogeneity along the HFR range cells is assumed, in-situ instruments provide point measurements); iii) vertical mismatch (adjustment of wind measurement from 10 m above mean sea level to sea surface). Other physical effects such as the wind duration and fetch should also be included in the evaluation of the HFR inversion performance.*"

References:

Shen, W.; Gurgel, K.-W. Wind direction inversion from narrow-beam HF Radar backscatter signals in low and high wind conditions at different radar frequencies. Remote. Sens. 2018, 10, 1480.

Lipa, B.; Barrick, D.; Alonso-Martirena, A.; Fernandes, M.; Ferrer, M.I.; Nyden, B. Brahan project high frequency radar ocean measurements: Currents, winds, waves and their interactions. Remote. Sens. 2014, 6, 12094–12117.

**Line 599 replace know-how with operational experience**

Done.

**Line 600 replace minor with reduced**

Done.

**Line 605 Is Med HFR group working with Ocean Best Practices https://www.oceanbestpractices.org ?**

Yes, it is. Scientists, technicians and operators from several Mediterranean institutions contribute to the development of OBPs in the context of different projects (e.g. JericoNEXT, JericoS3, EuroSea, etc). Indeed, by introducing the keywords "Mediterranean HF radar" in the Searching area of the OBP repository, the reviewer will be able to access documentation provided by many of the co-authors of this manuscript.

To our knowledge, the next upload will be in March, concerning a "Data Management Best practises report for physical mature platforms"', being developed in the context of the Jerico-S3 project.

The following paragraph has been added to the text:

"*In this context, it is worth mentioning that the Mediterranean HFR community is also actively working with Ocean Best Practices (https://www.oceanbestpractices.org), a global, sustained system comprising technological solutions and community approaches to enhance management of methods as well as support the development of ocean best practices.*"

**Line 608 does the numbering in the list denote a hierarchy? if not replace with symbols.**

Replaced with symbols.

**Line 617 is there a reference for the EuroGOOS HFR Task Team?**

There is a website: https://eurogoos.eu/high-frequency-radar-task-team/

One reference, available at https://hal.archives-ouvertes.fr/hal-03328829/, has been added:

Lorenzo Corgnati, Carlo Mantovani, Anna Rubio, Emma Reyes, Paz Rotllan, et al.. the eurogoos high frequency radar task team: a success story of collaboration to be kept alive and made growing. 9th EuroGOOS International conference, Shom; Ifremer; EuroGOOS AISBL, May 2021, Brest, France. pp.467-474.

**Line 622 consider removing "for the development of operational ocean monitoring via HFR systems"**

Done.

**Line 625 spell out EU, is there a reference for JERICO-NEXT?**

Done. Although there is no reference for JERICO-NEXT, there is a web link that has been added to the text: https://www.jerico-ri.eu/previous-project/jerico-next/

**Line 627 replace within with "with"**

Done.

**Line 631 which set the basis**

Done.

**Line 635 remove "the achievement of"**

Done.

**Line 637 can the authors comment how the European standard format compares to the American standard? Are there any differences?**

The paragraph has been moved to sections 2.5 and 2.6, following reviewer 2 suggestion.

The European common data and metadata model for NRT data has been implemented according to the standards of Open Geospatial Consortium (OGC) for access and delivery of geospatial data, and compliant with the Climate and Forecast Metadata Convention CF-1.6, to the Unidata NetCDF Attribute Convention for Data Discovery (ACDD), to the OceanSITES convention and to the INSPIRE directive.

Furthermore, it has been defined following the guidelines of the DATAMEQ working group and it fulfils the recommendations given by the Radiowave Operators Working Group (ROWG).

To enforce semantics and interoperability, controlled vocabularies are used in the model for variable short names and standard names.

The model specifies the file format (i.e. netCDF-4 classic model), the global attribute scheme, the dimensions, the coordinate, data and Quality Control (QC) variables and their syntax, the QC procedures and the flagging policy for both radial and total data.

However, discrepancies have been found between the European and the US common data and metadata format regarding: i) variables names and syntax (e.g. LONGITUDE; LATITUDE; EQCT; NSCTM GDOP in Europe vs. lon, lat, u, v, DOPx, DOPy in US ); ii) dimensions and coordinates (e.g. TIME:units = "days since 1950-01-01T00:00:00Z" in Europe vs. time:units = "seconds since 1970-01-01" in US); iii) global attributes (different or missing mandatory ones).

A piece of text has been added to the text to clarify this point:

"*Some discrepancies have been found between the European and the US common data and metadata format regarding: i) variables names and syntax; ii) dimensions and time coordinates; iii) global attributes (different or missing mandatory ones)."*

**Line 638 spell out or explain SDC and INSPIRE**

Done. SDC and INSPIRE stand for "supplemental digital content" and "infrastructure for spatial information in Europe", respectively. The paragraph has been moved to sections 2.5 and 2.6, following reviewer 2 suggestion.

**Line 639 replace "to be mandatorily" with "are consistently applied to HFR data as defined by the EuroGOOS"**

Done.

**Line 646 is there a link for the EU HFR Node that can be provided**

We have added the following link:

*"The EU HFR Node (Fig. 7) was established in 2018 by AZTI, CNR-ISMAR and SOCIB, under the coordination of the EuroGOOS HFR Task Team (Rubio et al., 2017), as the focal point and operational asset in Europe for HFR data management and dissemination (http://150.145.136.27:8080/thredds/HF_RADAR/HFradar_catalog.html) by promoting networking between EU infrastructures, marine data portals and the Global HFR network."*

**Line 660 can anything be said about data users, statistics on data use?**

We are still unable to provide the usage metrics at regional level. CMEMS InSitu TAC (one of the main European marine data portals providing standard surface currents derived from HFR) is working on the providers catalogue where every data provider will be able to check the number of users per platform. A virtual access metric system is also being developed in the context of Jerico S3 project and some of the data providers are already generating metrics on HFR data usage, but not at regional level.

**Line 668 does this rule apply to only HFR or all ocean measurements? please explain**

Only to HFR measurements, as indicated now in the text. This was the rule established by the European HFR node, considering that the implementation of the standards can often be beyond the technical capabilities of many scientific communities and, if not, these time-demanding tasks are usually carried out on a voluntary basis from the data provider, being typically funded by science activities.

**Line 697 should the items in the list be capitalised?**

Not really, items changed.

**Line 740 replace inexistent with nonexistent**

Done.

**Line 745 replace Served as with For**

Done.

**Table 2 spell out PdE and MIO, consider adding a column for the country and removing country from the Institution column. Should there be an order for the table? Alphabetical by country? frequency? Authors can decide.**

Done. A new column has been inserted into the table. The elements of the table have been ordered by frequency.

**Figure 8c should Phased array be replaced with beam forming if comparing to direction finding?**

Corrected. See below the new Figure 8:

[Figure]

**Figure 9b the 14(23%) is cutoff at the bottom, please correct**

Corrected. See below the new Figure 9:

[Figure]

**Line 785 serve the marine industry**

Done.

**Table 3 INTERREG is mentioned often, can this be explained.**

Yes, it has been explained in the text (as Table 3 has been replaced by a Gantt diagram, following the reviewer´s advice):

"*Interreg programmes are European Territorial Cooperation programmes, designed to promote cooperation between member states on shared challenges and opportunities*."

**Would the authors consider replacing the table with a Gantt chart to show the information?https://pythonawesome.com/a-convenient-but-aesthetic-way-of-creating-agantt-chart-with-python/**

Of course, we are willing to replace Table 3 with a Gantt chart, which is a user-friendly way to summarise the timeline of the projects focused on HFR technology in the Mediterranean.

To this aim, the information about the funding body has been moved to the section where the projects are thoroughly described, while the web links have been inserted into the Gantt diagram (new Figure 10, attached below), which contains the official logo of each project along with the temporal period covered by them.

[Figure]

**Line 792 the first two project have the acronym explained in the title, the remaining ones do not. Please be consistent.**

Fully agree. The acronyms have been explained in the main body of the manuscript, not in the title.

**Line 840 remove "which are also" and "in this section"**

Removed.

**Line 850 The project's overall objective was**

Corrected.

**Line 890 in title it is CALYPSO-FollowOn, on the next line the – is missing? Also missing years of the project**

Corrected.

**Line 910 can you add a summary sentence or paragraph on this section, something to tie together all the projects?**

Done. A summary paragraph has been added:

*"In conclusion, the last 10-15 years have witnessed the significant increase of national and cross-border projects in the Mediterranean Sea (Fig. 10) whose main scope was (and still is) to consolidate the HFR as an efficient coastal ocean monitoring technology. Most of the projects are funded by the European Commission in the framework of different Interreg programs, by the EU's H2020 Research and Innovation and by national research programs. In particular, 2020 has been a key year in terms of wealth of initiatives carried out simultaneously (9). A relevant number of new HFR sites have been recently deployed and integrated into multi-platform observatories, providing quality-controlled data that are routinely delivered to a broad audience and subsequently used for diverse marine applications, among others: maritime safety, oil spill accidents or SAR operations (TOSCA, HAZADR, IBISAR, SICOMARplus, CALYPSO, PANORAMED, SHAREMED, i-WaveNET), port and harbor security (SINAPSI), risk prevention and coastal management (TRADE), as well as marine spatial planning and integrated coastal zone management (RITMARE, IMPACT)."*

**Line 913 Equal to other**

Done.

**Line 915 remove "not only"**

Done.

**Line 916 move "strengths, weaknesses, opportunities and threats" to beginning of sentence. remove potential.**

Done

**Line 920 pursuit of permanent funding like those of other programs?**

We have expanded the sentence:

*"the pursuit of permanent funding (thanks to Interreg programs like SICOMAR-Plus or CALYPSO) to extend the network at both national and regional scales for better cross-border coverage."*

**Line 930 what kind of licenses? radio frequency? equipment installation? please state**

Clarified in the text: "*licences to source site permissions, site access, transmit licences and use of the data (Mantovani et al., 2020).*"

**Line 950 For this paragraph, is this process happening for other measurements? if so please list.**

We guess so, but do not have specific information about other observational platforms in the Mediterranean Sea. We are aware that, in the case of the Copernicus Marine Service (CMEMS), some Monitoring and Forecasting centres (MFCs) have delivered new 3D hourly forecast products (of physical variables like temperature, salinity and currents) as a result of the strong demand from a variety of end-users and intermediate-users that want to nest their coastal models into the CMEMS regional models. Previously, only daily means were available for the entire water column and the dynamical nesting was not so consistent.

We are not sure about the convenience of mentioning here the bidirectional commitment in CMEMS, but are open to include it in the manuscript if the reviewer wants so.

**Line 957 replace happy-ending with success**

Done.

**Line 969 Additionally pilot**

Done.

**Line 982 to mitigate noise and clutter**

Done.

**Line 988 can easily be detected, triggering alerts for operators**

Done.

**Line 989 spell out AIS, reference https://hfradar.msi.ucsb.edu/brian emery/files/reports/2013_sbir_phase2_final_report.pdf**

The acronym AIS stands for "Automatic Identification System". Added to the text along with the reference Whelan et al. (2013):

Whelan, C., Teague, C., Barrick, D., Emery, B. and Washburn, L.: HF Radar Calibration with Automatic Identification System Ships of Opportunity, Phase II Final Report, NOAA SBIR, Silver Spring, MD, USA, 2013.

**Line 1016 replace "keep down" with "reduce"**

Done.

**Line 1046 remove very**

Done.

**Line 1084 replace "build-up" with utilize**

Done.

**Line 1085 what is meant by first order, can this be further explained?**

We intended to highlight that it is of the greatest importance, significance, or magnitude. We have replaced "first-order" by "significant" to avoid any misunderstanding.

**Line 1092 examples illustrated here might**

We have provided an illustrative example: "such as the HFR network operating in Asia, presented at the 1st Ocean Radar Conference for Asia (ORCA) (Fujii et al., 2013)".

Where the reference is:

Fujii, S., Heron, M.L., Kim, K., Lai, J.W., Lee, S.H., Wu, X., Wyatt, L.R. and Yang, W.C: An overview of developments and applications of oceanographic radar networks in Asia and oceania countries, Ocean Science Journal, 48 (1), 69-97, 2013.

**REVIEWER #2:**

**General Comments**

**This paper is an ambitious community work aiming to showcase the current status of the Mediterranean HFR network and the future roadmap for coordinated actions that will allow this to play a major role in the high-level challenges of the ocean observing landscape in the Mediterranean Sea. Significant innovations are described, with interesting multi-site approaches and covering a very wide spectra of fields in the overall value-chain from the HFR systems operations to the transfer of advanced data products. The presented work is also gathering a complete review of the main levers that the**

**community is tackling (BPs, Harmonization, Data Quality, New parameters…) for promoting exchanges between operators, and creating synergies and added value by transforming a set of individual radars into an integrated network. The description of the community status, difficulties, key references and challenges derives to a very useful roadmap for the current actors of the network, also for the potential future contributors, and in general for the ocean observing community.**

**The established regional roadmap is well linked to the European and Global initiatives. Some regional specificities are well described, in particular in the SWOT analysis. However, it may be clarified which of those challenges for future development is really answering a specific or prioritised issue for the Region, and which are shared with the European or Global community. The manuscript will definitely represent an important step forward for the ocean observing community. Some detailed minor changes and recommendations for improving the manuscript are listed below:**

Many thanks to the reviewer for the number of useful comments that will help to significantly improve the quality of the final version of this manuscript. In particular, section 5 of the document has been upgraded to better clarify:

*"The Mediterranean HFR network must face a variety of challenges for future development, most of which are shared with the European and Global HFR communities. However, few other aspects are specific to the Mediterranean Sea due to the intrinsic peculiarities of this regional semi-enclosed basin. For instance, the presence of reflections from moving ships or radio frequency interferences from (non)official radio services are more pronounced as the maritime traffic is significantly intense in coastal Mediterranean areas (Bellomo et al., 2015). In this context, obtaining dedicated frequency allocation for HFR technology remains the top priority issue. A network extension to cover a substantial portion of the Mediterranean coastline also constitutes a prime concern, especially in the southern shore countries where the monitoring capabilities are extremely limited. This process is handicapped by the prominent use of medium and short range HFR systems which map smaller spatial domains. Moreover, the Strait of Gibraltar, the Dardanelles and the Suez Canal (Fig. 1) act as physical constraints, leading to slow water renewal cycles and high residence times at basin scale. Consequently, global warming and the chronic degree of pollution related to anthropogenic activities impose higher pressures than in any other sea in the world, turning the Mediterranean Sea into a more vulnerable hot spot for climate change (Tuel and Eltahir, 2020)."*

**l.138 General capabilities of HFR are explained, but here or elsewhere in the paper, there is no mention to the limitations of using Long Range in semi-enclosed seas like the Mediterranean Sea (difference with the IBIROOS area where there is a significant number of Long-Range systems). This can be also mentioned later in the SWOT Weaknesses or at least as a factor for achieving the full coverage.**

In the SWOT diagram (weaknesses section), we have replaced "Still limited coverage in the Mediterranean Sea" by "*Limited coverage in the Mediterranean Sea, exacerbated by the predominant use of medium/short range HFR systems.*"

Equally, we have added the following piece of text at the beginning of section 3:

*"The Mediterranean HFR network includes 15 different systems, which cover a small portion of the entire coastal domain (Fig. 1). The limited spatial coverage is not only due to the reduced number of HFR deployed but also to the predominant use of medium (13.5 MHz) and short (above 20 MHz) range systems, whose basic technical aspects are gathered in Table 2. While these HFRs present a maximum range of 80 km, long range systems (which operate below 5 MHz and are typically deployed in the Atlantic European waters) can map the surface circulation over broader areas for distances up to 200 km offshore."*

Moreover, we have also added a mention to the limitations of using long-range in semi-enclosed seas like the Mediterranean Sea (at the beginning of section 3 of the manuscript):

*"Long-range HFR systems are not deployed in the Mediterranean since they present some technical limitations in this semi-enclosed sea. On one hand, they provide surface circulation maps with coarser horizontal grid resolution (above 5 km), which are not convenient to adequately resolve some (sub)mesoscale ocean processes (i.e., eddies, instabilities, etc.) that commonly characterise the Mediterranean dynamics. On the other hand, they cannot accurately monitor the wave field under low sea states as the second-order spectrum is closer to the noise floor (and more likely to be contaminated with spurious contributions) than in the case of short and medium range HFR systems. As the Mediterranean wave climate is not as intense as the Atlantic one, the use of long-range systems would result in limited precision and reduced temporal continuity in wave measurements (Lipa and Nyden, 2005). Finally, it is worth mentioning that a cross-border agreement was signed in 2018 (by in Spain, France and Italy) to establish the 13-16 MHz band as the one to be used for oceanographic radars in the Western Mediterranean Sea (Roarty et al., 2019)".*

**Fig.2 A Mediterranean HFR Working Group is mentioned. In the text l.184, it is called Mediterranean HFR network. For consistency, this would be named in Fig2 (not to be confused with Observation Working Group, one of the 3 MONGOOS WGs).**

Apologies, there was a typo in Figure 2. We wanted to mean "Mediterranean HFR Task Team". Actually we think that there is a subtle but significant difference between both concepts, the Mediterranean HFR Task Team (belonging to the Observing working Group from MONGOOS) and the Mediterranean HFR network. The former is a representation of the latter. While the Mediterranean HFR network comprises all the HFR systems deployed in this regional sea and the respective operators, the Mediterranean HFR Task Team constitutes a smaller entity that represents and coordinates joint actions and research contributions from the entire network. Indeed, some members of the Mediterranean network are not involved in the task team (for instance, the HFR-Dardanos system operated by our Greek colleagues). Other institutions (like those operating HFR-Israel) are open to collaborate in specific initiatives (like the present paper) but, again, are not active participants of the Mediterranean HFR task team.

**l.200 Errata: the EuroGOOS HFR Task Team (word order and upper case for Task)**

Corrected!

**l.237: It may be more precise to say 0.5-5m in "operating at specific frequencies within the 3-30 MHz band and providing radial measurements which are representative of current velocities in the upper 0.5–2 m of the water column. See Rubio et al. 2017**

OK! As indicated in Rubio et al. (2017), the maximum integration depth is 420 cm so we have rounded and simplified to: 0.5-4 m. A reference to Rubio et al. (2017) has been also added.

**In 2.1 Fundamentals of HFR technology, it could have been mentioned how the common sea states of a semi-enclosed sea like Med may impact the performance of longer range HFR, with possible consequences to be taken into account in the plan for achieving a full coverage of the Mediterranean coastline.**

Fully agree. As previously indicated, we have mentioned the limitations of using long-range in semi-enclosed seas like the Mediterranean Sea at the beginning of section 3 of the manuscript (instead of section 2.1):

*"Long-range HFR systems are not deployed in the Mediterranean since they present some technical limitations in this semi-enclosed. On one hand, they provide surface circulation maps with coarser horizontal grid resolution (above 5 km), which are not convenient to adequately resolve some (sub)mesoscale ocean processes (i.e., eddies, instabilities, etc.) that commonly characterise the Mediterranean dynamics. On the other hand, they cannot accurately monitor the wave field under low sea states as the second-order spectrum is closer to the noise floor (and more likely to be contaminated with spurious contributions) than in the case of short and medium range HFR systems. Since the Mediterranean wave climate is not as intense as the Atlantic one, the use of long-range systems would result in limited precision and reduced temporal continuity in wave measurements (Lipa and Nyden, 2005). Finally, it is worth mentioning that a cross-border agreement was signed in 2018 (by in Spain, France and Italy) to establish the 13-16 MHz band as the one to be used for oceanographic radars in the Western Mediterranean Sea (Roarty et al., 2019)"*

**l.426. The abbreviation GoN is used before the full version that appears l.456.**

Corrected in line 426 and also in Table 1.

**Table 1. For consistency, I would recommend to use "Gulf of Naples" in Table 1.**

Done.

**l.448: Long et al 2011 (Central California) should not be included in references on European waters.**

True! Deleted from the list of references.

**l.459: using two "alternative data sources" rather than "different platforms"**

Done!

**l.465 errata: against**

Corrected.

**Figure 5a. Quality of graphs should be improved (size of labels, same limits for Axes Y, same type of lines, general image sharpness).**

Improved.

**l.569: Add a comma: In the Ligurian Sea experiment,**

Added.

**l.571: adda comma: During the experiment,**

Added.

**Section 2.4: the beginning of Section 2.4 appears too general for this section and may better fit in section 1.5. Only aspects dealing with data flow may be kept in the historical review of the roles of the different initiatives.**

Fully agree. The first paragraph has been shortened and reformulated, following the reviewer's suggestion. The piece of text eliminated has not been integrated into 1.5 since it is a little bit redundant with other paragraphs along the manuscript.

**L636-642: The text may be simplified here as it is a bit redundant with what is explained later in 2.5 and 2.6.**

Fully agree. The following paragraph has been deleted from L636-L642 and integrated into sections 2.5 and 2.6 to avoid any redundancy:

*"The European standard format for HFR data and metadata model has been defined and implemented, compliant with Climate and Forecast Metadata Convention version 1.6 (CF-1.6), OceanSITES convention, CMEMS-INSTAC and SDC requirements and INSPIRE directive.*

*Furthermore, a battery of the QC tests to be mandatorily applied to HFR data has been defined 640 according to the EuroGOOS Data Management, Exchange and Quality Work Group (DATAMEQ) working recommendations on real-time QC and building on the Quality Assurance/Quality Control of Real-Time Oceanographic Data (QARTOD) manual produced by the US Integrated Ocean Observing System (IOOS)"*

**l.651 Figures by mid-2021 should be presented as a result, not as an objective. Or the date and corresponding objective should be updated.**

Right. The sentence has been modified:

*"Within the European framework, the EU HFR Node is currently managing data from 16 systems
(http://150.145.136.27:8080/thredds/HF_RADAR/HFradar_CMEMS_INSTAC_catalog.html) . In particular, 5 of these 16 systems (31%) are deployed in the Mediterranean coastline and belong to the MONGOOS network: HFR-Gibraltar, HFR-Ibiza, HFR-DeltaEbro, HFR-TirLig and HFR-NAdr (Fig. 1)."*

**4. Multi-institutional collaborative projects with HFRs in the Mediterranean Sea: I would suggest to mention the ongoing JERICO-S3 and JERICO-DS projects, part of the JERICO-RI initiative. Their impact could be significant in terms of integration of HFR among key coastal observing technologies.**

Done! We have added the following paragraph:

"In this context, it is worth mentioning the ongoing JERICO-S3 and JERICO Design Study (DS) projects (2020-2023), as part of the JERICO Research Infrastructure (RI) initiative. JERICO-RI, which is a long-term integrated framework providing high-quality marine data, expertise and multi-platform infrastructures for Europe's coastal seas, might have a significant impact in terms of integration of HFR among key coastal observing technologies."

**l.923: I would suggest expressing differently the reason for a clear north-south unbalance in the Mediterranean region. The MOONGOOS community knows better than me the regional variability. More than the Political systems themselves, the factors may lie in the differences between environmental policies, resources dedicated to marine monitoring and research programs, socioeconomical and political priorities, etc.**

Fully agree. The sentence has been expanded to better clarify the reasons:

*"Furthermore, the monitoring capabilities are variable, with a clear north-south unbalance in the Mediterranean region due to a variety of reasons. In addition to the existence of fragile and volatile political systems in southern shore countries that seriously handicap sustained research programs (Fig. 11), precarious socio-economic conditions also impact on the political priorities. Intermittent and uncoordinated initiatives might result in underdeveloped marine policies (at both national and regional level), significant resources dispersion and the inefficient management of the coastal environment. In this context, the implementation of lower-cost HFRs would greatly enhance developing countries' capability to monitor coastal*

*waters and to establish new alliances and regional partnerships. The link between MONGOOS and GOOS Africa must be strengthened in order to define common roles and shared activities in the Mediterranean Sea."*

**Figure 10. the different Threats may be organized with "Insufficient adoption of HFR currents standardization" after other linked issues like "Lack of agreement on the data policy".**

The section "Threads" of the SWOT diagram (Fig. 11) has been reorganised and grouped into 2 different categories to better clarify the risks associated with the Mediterranean HFR network.

[Figure]

**In 5. Future challenges and prospects:**

**As part of the roadmap to transform individual radars into an integrated network, a phased approach is proposed. The different steps aim to optimize and consolidate the network of existing systems. However, the network is currently covering "a small portion of the entire coastal domain". As part of this roadmap, authors could add plans for:**

**- Defining a quantitative objective (number of systems, surface covered…) as a long-term target**

Considering that only 2% of the world's coastline is currently monitored with HFRs (Moltmann et al., 2019) and the 46000 km of coastline length in the Mediterranean, is would be recommended to progressively increase the number of HFR sites, keeping the European rate of 6 new HFR sites installed per year (Rubio et al., 2017), particularly in critical data-sparse areas that prove challenging for other observation platforms, with a clear focus on outcomes and societal benefit.

In particular, we have added the following paragraph in section 5.1.:

*"As a quantitative long-term objective, it would be recommended to maintain the rate reported in Rubio et al. (2017) of 6 new HFR sites installed per year in Europe. That might imply the installation of 2-3 new HFR sites per year in geostrategic coastal regions of the Mediterranean Sea such as marine protected areas, straits or port-approach areas."*

Reference

Moltmann, T., Turton, J., Zhang, H.-M., Nolan, G., Gouldman, C., Griesbauer, L., Willis, Z., Piniella, Á. M., Barrell, S., Andersson, E., Gallage, C., Charpentier, E., Belbeoch, M., Poli, P., Rea, A., Burger, E. F., Legler, D. M., Lumpkin, R., Meinig, C., … Zhang, Y. (2019). A Global Ocean Observing System (GOOS), Delivered Through Enhanced Collaboration Across Regions, Communities, and New Technologies . In Frontiers in Marine Science (Vol. 6). https://www.frontiersin.org/article/10.3389/fmars.2019.00291

**- Agreeing a joint methodology to define priority areas at regional level in the development of the network (for example as introduced in JERICO-NEXT Deliverable D3.4)**

Following the methodological guidelines to define the joint strategy to design an integrated HFR network at regional scale developed in the context of Jerico Next project (Griffa et al., 2019), a combination of societal needs and HFR technology limitations should be considered within this purpose. Regarding the former and taking into account the importance of the HFR data and applications for Maritime Safety, the joint analysis of the marine traffic density maps (by using AIS data), historical SAR incidents (reported by all the coastal countries), location of bunkering operation areas and the location of migratory routes in the Mediterranean Sea can help to design the geographical distribution of the future installations.

Moreover, the study of the environmental sensitivity of the coastline, categorised based on the geomorphological classification of the coast, the biological resources (coastal protected areas, fish recruitment, areas of dispersion and retention of larvae, etc) and the human use (i.e. infrastructures, services, cultural and historic resources), as defined by NOAA (2002), is also recommended for the mapping of monitoring needs.

The improvement of models through HFRr data assimilation in potential locations can also be assessed based on OSE/OSSEs methods, thus contributing to better design the network expansion.

In addition, the wave climate must be considered for selecting the HFRs central operating frequency (i.e. limiting the use of long-range in the Mediterranean), the shape of the coastline will impact the GDOP error and should be contemplated, and the site locations must comply with the separation distances as advised by the ITU (Mantovani et al., 2020) and with the recommendations to mitigate wind turbine interference impacts on HFR (Kirincich et al., 2019)

References:

1) Petersen, J., et al. 2019. Environmental Sensitivity Index Guidelines, Version 4.0. NOAA Technical Memorandum NOS OR&R 52.

2) Griffa, A.; Horstmann, J. and Mader, J. et al (2019) Report on final assessment of methodological improvements and testing. JERICO-NEXT WP3 Innovations in Technology and Methodology, Deliverable D3.4, Version 2. Brest, France, IFREMER, 56pp. (JERICO-NEXT-WP3-D3.4-180719-V2). DOI:http://dx.doi.org/10.25607/OBP-948

Finally, the following paragraph has been added to section 5.1:

*"To better define priority installation areas at regional level, methodological guidelines were developed in the context of JERICO-NEXT project (Griffa et al., 2019), where a combination of societal needs (maritime traffic density, historical SAR incidents, location of bunkering areas, biological resources, etc.) and HFR technology limitations were jointly considered. Similarly, the Mediterranean HFR network should be further implemented following these shared guidelines."*

**- Performing coordinated actions towards stakeholders (for example towards National GOOS Focal points who are serving in EOOS Operation Committee)**

The following paragraph has been added to section 5.1.:

*"Potential stakeholders should be clearly identified and promptly informed to boost their engagement. Coordinated actions to involve the national focal points (which are the appropriate contact points in each member state for affairs regarding the implementation of the GOOS at national and global levels) should be also performed within the European Ocean Observing System (EOOS) framework".*

**Additionally, all the general, technical and Research aspects are relevant for the Med and beyond at European and Global level. But it would be interesting to mention a bit more explicitly how will be tackled some regional specificities exposed in the SWOT analysis.**

Done!

Regarding the "north-south unbalance" in the monitoring capabilities (Figure 11):

*"Furthermore, the monitoring capabilities are variable, with a clear north-south unbalance in the Mediterranean region due to a variety of reasons. In addition to the existence of fragile and volatile political systems in southern shore countries that severely handicap sustained research programs (Fig. 11), precarious socio-economic conditions also impact on the political priorities. Intermittent and uncoordinated initiatives might result in underdeveloped marine policies (at both national and regional level), significant resources dispersion and the inefficient management of the coastal environment. In this context, the implementation of lower-cost HFRs would greatly enhance developing countries' capability to monitor coastal waters and to establish new alliances and regional partnerships. The link between MONGOOS and GOOS Africa must be strengthened in order to define common roles and shared activities in the Mediterranean Sea".*

Regarding the "limited training of technicians" (Figure 11):

*"training of new technicians to operate the network, which would include the dissemination of the latest available methodologies to ensure that the most up-to-date best practices are followed [..] holding open-house conferences and workshops, not only focused on HFR operator community and permitting agencies but also on a more general non-instructed audience, might be an effective way of promoting public awareness and ensuring the network's survival.*

Regarding the "difficulties for cross-border-agreements:"

*"In spite of the fruitful collaborations between the HFR national networks, the coordination and long-term integration at regional scale are sometimes handicapped by poor data policy and restricted data access (Fig. 11). There is still a recognized necessity for the unification of standards, the centralization of methodologies and best practices documentation to increase not only the interoperability of the coastal HFRs network design, operation and maintenance tasks but also the efficient data discovery (Mantovani et al., 2020). In this context, new cross-border agreements should be reached to consolidate the observing infrastructure, following the example of that one signed in 2018 by Spain, France and Italy in the Western Mediterranean Sea (Roarty et al., 2019). For instance, the surface circulation monitoring in the Strait of Gibraltar could be significantly enhanced thanks to a cross-border agreement between Spain and Morocco to integrate their respective HFR sites into one single network"*.

---

## Author Response (AR2)

Thanks a lot to the Editor for the kind suggestions.

Figure 10 has been updated according to the modifications asked by the reviewer 2.